# Frozen-soil hydrological modeling for a mountainous catchment at northeast of the Qinghai-Tibet Plateau

Hongkai Gao 1, 2, 3*, Chuntan Han 4, Rensheng Chen 4, Zijing Feng 2, Kang Wang 1,2, Fabrizio Fenicia 5, Hubert Savenije 6

Key Laboratory of Geographic Information Science (Ministry of Education of China), East China Normal University, Shanghai, China
School of Geographical Sciences, East China Normal University, Shanghai, China
State Key Laboratory of Tibetan Plateau Earth System and Resources Environment (TPESRE), Institute of Tibetan Plateau Research, Chinese Academy of Sciences, Beijing, China.
Qilian Alpine Ecology and Hydrology Research Station, Key Lab. of Ecohydrology of Inland River Basin, Northwest Institute of Eco-Environment and Resources, Chinese Academy of Sciences, Lanzhou 730000, China
Eawag, Swiss Federal Institute of Aquatic Science and Technology, Dubendorf, Switzerland
Delft University of Technology, Delft, the Netherlands

*Corresponding to: Hongkai Gao (hkgao@geo.ecnu.edu.cn; gaohongkai2005@126.com)

## Abstract:

Increased attention directed at frozen-soil hydrology has been prompted by climate change. In spite of an increasing number of field measurements and modeling studies, the impact of frozen-soil on hydrological processes at the catchment scale is still unclear. However, frozen-soil hydrology models have mostly been developed based on a "bottom-up" approach, i.e. by aggregating prior knowledge at pixel scale, which is an approach notoriously suffering from equifinality and data scarcity. Therefore, in this study, we explore the impact of frozen-soil at catchment-scale, following a "top-down" approach, implying: expert-driven data analysis → qualitative perceptual model → quantitative conceptual model → testing of model realism. The complex mountainous Hulu catchment, northeast of the Qinghai-Tibet Plateau (QTP), was selected as the study site. Firstly, we diagnosed the impact of frozen-soil on catchment hydrology, based on multi-source field observations, model discrepancy, and our expert knowledge. Two new typical hydrograph properties were identified: the low runoff in the early thawing season (LRET) and the discontinuous baseflow recession (DBR). Secondly, we developed a perceptual frozen-soil hydrological model, to

explain the LRET and DBR properties. Thirdly, based on the perceptual model and a landscape-based modeling framework (FLEX-Topo), a semi-distributed conceptual frozen-soil hydrological model (FLEX-Topo-FS) was developed. The results demonstrate that the FLEX-Topo-FS model can represent the effect of soil freeze/thaw processes on hydrologic connectivity and groundwater discharge and significantly improve hydrograph simulation, including the LRET and DBR events. Furthermore, its realism was confirmed by alternative multi-source and multi-scale observations, particularly the freezing and thawing front in the soil, the lower limit of permafrost, and the trends in groundwater level variation. To the best of our knowledge, this study is the first report of LRET and DBR processes in a mountainous frozen-soil catchment. The FLEX-Topo-FS model is a novel conceptual frozen-soil hydrological model, which represents these complex processes and has potential for wider use in the vast QTP and other cold mountainous regions.

# 1 Introduction

## 1.1 Frozen-soil hydrology: one of twenty-three unsolved problems

The Qinghai-Tibet Plateau (QTP) is largely covered by frozen soil and is characterized by a fragile cold and arid ecosystem (Immerzeel et al., 2010; Ding et al., 2020). As this region serves as the "water tower" for nearly 1.4 billion people, understanding the frozen soil hydrology is important for regional and downstream water resources management and ecosystem conservation. Frozen soil prevents vertical water flow which often leads to saturated soil conditions in continuous permafrost, while confining subsurface flow through perennially unfrozen zones in discontinuous permafrost (Walvoord and Kurylyk, 2016). As an aquiclude layer, frozen soil substantially controls surface runoff and its hydraulic connection with groundwater. The freeze–thaw cycle in the active layer significantly impacts soil water movement direction, velocity, storage capacity, and hydraulic conductivity (Bui et al., 2020; Gao et al., 2021).

Frozen-soil hydrology attracts increasing attention, as the cold regions, e.g. QTP and Arctic, are undergoing rapid changes (Song et al., 2020; Tananaev et al., 2020). Frozen-soil thawing also poses great threats to the release of frozen carbon in both high altitude and latitude regions, which is likely to create substantial impacts on the climate system (Wang et al., 2020). Attention is also growing for the impact of frozen-soil hydrology on nutrient transport and organic matter, and frozen soil–climate feedback (Tananaev et al., 2020). Hence, there are strong motivations to better understand frozen-soil hydrological processes (Bring et al., 2016).

Frozen-soil degradation and its impact on hydrology is one of the research frontiers for the hydrologic community (Blöschl et al., 2019; Zhao et al., 2020; Ding et al., 2020). "How will cold region runoff and groundwater change in a warmer climate?" was identified by the International Association of Hydrological Sciences (IAHS), as one of the 23 major unsolved

scientific problems (Blöschl et al., 2019), which requires stronger harmonization of
community efforts.

## 1.2 The frontier of frozen-soil hydrology

Knowledge on frozen-soil hydrology was acquired through detailed investigations at
isolated locations over various time spans by hydrologists and geocryologists (Woo et al.,
2012; Gao et al., 2021). At the core scale, there are many measurements of soil profiles,
including but not limited to soil temperature (Kurylyk et al., 2016; Han et al., 2018), soil
moisture (Dobinski, 2011; Chang et al., 2015), groundwater fluctuation (Ma et al., 2017;
Chiasson-Poirier et al., 2020), and active layer seasonal freeze-thaw processes (Wang et al.,
2016; Farquharson et al., 2019). At the plot/hillslope scale, land surface energy and water
fluxes are measured by eddy covariance, large aperture scintillometer (LAS), lysimeter, and
multi-layers meteorological measurements. Geophysical detection technology allows us to
measure various subsurface permafrost features. At the basin scale, except for traditional
water level and runoff gauging, water sampling and the measurements of isotopes and
chemistry components provide important complementary data to understand catchment
scale hydrological processes (Streletskiy et al., 2015; Ma et al., 2017; Yang et al., 2019).
Remote sensing technology, including optical, near- and thermal-infrared, passive and
active microwave remote sensing, has been used to identify surface landscape features (e.g.
vegetation and snow cover) and directly or indirectly retrieve subsurface variables (e.g.
near-surface soil freeze/thaw and permafrost state) in frozen-soil regions (Nitze et al., 2018;
Jiang et al., 2020).
Besides measurement, modeling provides another indispensable dimension to understand
frozen-soil hydrology in an integrated way, and make predictions in climate change. There
has been a revival in the development of frozen-soil hydrological models simulating
coupled heat and water transfer. Such physically-based models typically calculate seasonal
freeze–thaw through solving heat transfer equations. Such equations are either solved
analytically or numerically (Walvoord and Kurylyk, 2016). The Stefan equation is a typical
example of the analytical approach, which calculates the depth from the ground surface to
the thawing (freezing) horizon by the integral of ground surface temperature and soil
features. The Stefan equation is widely used to estimate active layer thickness (Zhang et al.,
2005; Xie and Gough, 2013), and is incorporated into some hydrological models (Wang L,
2010; Fabre et al. 2017). The numerical solution schemes (e.g., finite difference, finite
element, or finite volume) to model ground freezing and thawing, is typically applied to
one-dimensional infiltration into frozen soils, and is included in models such as SHAW (Liu
et al. 2013), CoupModel (Zhou et al., 2013), the distributed water-heat coupled (DWHC)
model (Chen et al. 2018), the distributed ecohydrological model (GBEHM) (Wang Y. 2018),
and the three-dimensional SUTRA model (Evans et al. 2018). Andresen et al (2020)
compared 8 permafrost models on soil moisture and hydrology projection across the major
Arctic river basins, and found that the projection varied strongly in magnitude and spatial
pattern. Except for hydrological models, many land surface models explicitly consider the
freeze-thaw process, in order to improve land surface water and energy budget estimation
and weather forecasting accuracy in frozen-soil areas. Such models include VIC (Cuo et al.,
2015), JULES (Chadburn et al., 2015), CLM (Niu et al., 2006; Oleson et al., 2013; Gao et al.,
2019), CoLM (Xiao et al., 2013), Noah-MP (Li et al., 2020), ORCHIDEE (Gouttevin et al.,
2012). Comprehensive reviews on frozen-soil hydrological models can be found in
Walwoord and Kurylyk (2016), Jiang et al. (2020), and Gao et al. (2021).

## 1.3 The challenge of frozen-soil hydrological modeling

Although numerous frozen-soil hydrological models were developed, most models have
strong prior assumptions on the impacts of frozen-soil on hydrological behavior (Walvoord
and Kurylyk, 2016; Gao et al., 2021). Such models follow a "bottom-up" modeling approach,
which presents an "upward" or "reductionist" philosophy, based on the aggregation of
small-scale processes and *a priori* perceptions (Jarvis, 1993; Sivapalan et al., 2003). However,
most of the "upward" process understanding has been obtained from in-situ observation
and in-situ modeling, which have limited spatial and invariably limited temporal coverage
(Brutsaert, and Hiyama, 2012). It is worthwhile to note that frozen-soil has tremendous
spatial-temporal heterogeneities, which are strongly influenced by many intertwined factors,
including but not limited to climate, topography, geology, soil texture, snow cover, and
vegetation. Upscaling could average out some variables, and turn other variables visible and
even become dominant processes (Fenicia and McDonnell, 2022). Unfortunately, translating
spot/hillslope scale frozen-soil process to its influence on catchment scale hydrology,
guided by carefully expert analysis, and constrained by multi-source measurements, is still
largely unexplored.
The effects of the soil freeze/thaw process on hydrology at catchment scale is still
inconclusive. In the headwaters of the Yellow River, some modeling studies concluded that
permafrost has significant impact on streamflow (Sun et al., 2020). But in Sweden and the
northeast of the United States, other studies found frozen soil have negligible impact on
streamflow (Shanley and Chalmers, 1999; Lindstrom et al., 2002). Some studies found that
the impact of frozen soil on streamflow is concentrated in certain periods. For example,
Osuch et al. (2019) found permafrost to impact on groundwater recession and storage
capacity of the active layer in Svalbard Island; Nyberg et al. (2001) found that in the Vindeln
Research Forest in northern Sweden permafrost impacted streamflow only in springs.
Hence, we argue that the impact of local scale freeze-thaw process on runoff should be
regarded as a hypothesis to be verified or rejected.
The unexplored frozen-soil hydrology is especially true for mountainous Asia, due to the
lack of long-term observations as a result of the difficulty of access and high cost of
operation. The cold region of the QTP is characterized by relatively thin and warm frozen-
soil with low ice content, due to the unique environmental conditions, arid climate, high
elevation and steep geothermal gradient (Cao et al., 2019; Zhao et al., 2020; Jiang et al.,
2020). Snow cover is thinner, and vegetation cover is poorer than in Arctic regions. These
features limit the insulation effect on freeze-thaw processes, resulting in a much larger
active layer depth (Pan et al., 2016). Topographical features, including elevation and aspect,
are major factors affecting permafrost distribution. The complex mountainous terrain, as a

result of recent tectonic movement, leads to large spatial heterogeneity in the energy and water balance, and underexplored frozen-soil hydrology on the QTP (Gao et al., 2021).

## 1.4 Aims and scope

In this study, we utilized a "top-down" approach (Sivapalan et al., 2003), to understand the effect of frozen-soil on hydrology in the Hulu catchment on the northeastern edge of the QTP. The aims of this study are as below:

1) Diagnosing the impacts of frozen-soil on hydrology in the mountainous Hulu catchment, with multi-source, multi-scale data and model discrepancy;

2) Developing a quantitative conceptual frozen-soil hydrological model, based on expert-driven interpretation in the form of perceptual model for the Hulu catchment;

3) Testing the realism of the conceptual frozen-soil hydrological model, with multi-source and multi-scale observations.

In this paper, we firstly introduced the study site and data in Section 2; an expert-driven perceptual frozen-soil hydrology model was proposed in Section 3; a semi-distributed conceptual frozen-soil hydrological model, FLEX-Topo-FS, was developed in Section 4; the realism of the FLEX-Topo-FS model was tested in Section 5; in Section 6 and 7, last but not least, we made discussions and draw the conclusions.

## 2 Study site and data

The Hulu catchment (38°12′–38°17′ N, 99°50′–99°54′E) is located in the upper reaches of the Heihe River basin, the northeast edge of the QTP in northwest China (Figure 1). The elevation ranges from 2960 to 4820 m a.s.l., gradually increasing from north to south (Figure 1) (Chen et al., 2014; Han et al., 2018). Most precipitation occurs in the summer monsoon time, and snowfall in winter is limited (Han et al., 2018; Jiang et al., 2020). There is a runoff gauging station at the outlet, controlling an area of 23.1 km$^2$. Two minor tributaries are sourced from glaciers (east) and moraine–talus (west) zones, which merge at the catchment outlet. The Hulu catchment has rugged terrain and very little human disturbance. We identified four main landscape types, i.e. glaciers (5.6%), alpine desert (53.5%), vegetation hillslope (37.5%), and riparian zone (3.4%) (Figure 2).

The Hulu catchment mostly extends on seasonal frozen-soil and permafrost (Zou et al., 2014; Ma et al., 2021). Field survey in the upper Heihe revealed that the lower limit of permafrost was 3650m~3700m (Wang et al., 2013), above that elevation is permafrost, and below is seasonal frozen-soil. In the Hulu catchment, the lower limit of permafrost is around 3650m (Figure 1). Permafrost covers 64% of the catchment area, and the seasonal frozen-soil covers 36%. There is a strong co-existence between soil freeze-thaw feature and landscapes (Figure 1). Permafrost and moraine/talus with poor vegetation cover co-exist in higher elevation, with large heat conductivity and less heat insulation in winter, resulting deep frozen depth. The seasonal frozen soil in relative lower elevation has better vegetation

cover, with better heat insulation and less heat conductivity in winter, resulting in shallower frozen depth.

The elevation of the hydrometeorological gauging station is 2980m. We collected daily runoff, daily average 2m air temperature, and daily precipitation from January 1$^{st}$ 2011 to December 31$^{st}$ 2014. There was a flood event in 2013, which damaged the water level sensor, resulted in a runoff data gap from June 17$^{th}$ to July 10$^{th}$ in 2013. Soil moisture was measured in 20cm, 40cm, 80cm, 120cm, 180cm, 240cm, and 300cm depths from October 1$^{st}$ 2011 to December 31$^{st}$ 2013, with a data gap between August 3$^{rd}$ 2012 and October 2$^{nd}$ 2012. In the same soil moisture site, we also observed the soil freeze/thaw depth from 2011 to 2014. Groundwater depth was measured at WW01 site by four wells, with depth of 5m, 10m, 15m, and 25m respectively (Pan et al., 2021). The observation period was from 2016 to 2019, not overlaid with other hydrometeorological variables. Hence, we merely used the groundwater level to qualitatively constrain and test our perceptual model.

# 3 An expert-driven perceptual frozen-soil hydrology model based on field observations

Perceptual model is increasingly recognized as the central importance in hydrological model development (Fenicia and McDonnell, 2022). Although perceptual model is a qualitative representation of hydrological system, it bridges the gap between experimentalists and modelers, and hold the base for quantitative conceptual model. In this study, we developed a perceptual frozen-soil hydrology model for the Hulu catchment, based on field measurements and our expertise.

## 3.1 Observation1: Low runoff in the early thawing season (LRET)

Precipitation-runoff time series analysis is a tradition and powerful tool to understand catchment hydrology. When we plot Hulu catchment's precipitation and runoff data together, we observed an interesting phenomenon, i.e. low runoff in the early thawing season (LRET) (Figure 3). For example, on June 5-9, 2013, there was a 45.7mm rainfall event, with air temperature ranging from 3.0℃ to 11.9℃, but with only 0.68mm in total runoff generation. Ma et al., (2021) also showed that in the warm middle June 2015 in Hulu catchment, there was a large rainfall event (over 30mm/d), but no runoff response was observed. Moreover, this LRET phenomenon repeatedly happens every year, which allows us to exclude the possibility of measurement errors.

To further investigate the LRET phenomenon, we plot the time series of observed daily precipitation, temperature, runoff, freeze/thaw front depth, and soil moisture profile (20cm, 120cm, and 240cm) together (Figure 3). Soil moisture data showed that top soil was dry at the beginning of thawing season. Gradually, soil was thawing from both topsoil and downwards (Figure 3), and simultaneously the soil moisture was increased also downwards.

Although the topsoil was thawed, there was still frozen-soil underneath (Figure 3). The
water above the frozen-soil layer was even saturated with ponding but no percolation and
runoff generation. Moreover, the groundwater level further declined (Figure 11). This
illustrated that, during this period, the soil and groundwater system were disconnected, very
likely because the frozen soil blocked the percolation. As revealed by isotope data in the
Hulu catchment, groundwater contributed the dominant streamflow, i.e. 95% during the
frozen period (Ma et al., 2021). Thus, during this process, there was almost no runoff
generation, and the only contribution to streamflow during this period was the discharge
from groundwater system as baseflow. Our field work experience also verified that in the
early thawing season, vehicles were easy to be trapped in mires, because the surface soil
was muddy, saturated even over-saturated with ponding. When frozen-soil was completely
thawed in summer, trafficability became much better.
When completely thawed, the bidirectional thaw fronts meet, the soil moisture in the
bottom of frozen soil (around 2.4m in this study site, Figure 3) was increased sharply, and
then decreased, with a short period pulse. We also noted that the observation sites of
freeze/thaw front depth and soil moisture were both near the outlet of the Hulu catchment
in lower elevation (Figure 1). This means once the frozen soil in lower elevation was thawed,
soil and groundwater systems were reconnected. Hydrological processes, including
groundwater percolation and runoff generation, became the same as free of frozen-soil
normal circumstances.
**Learning from paired catchments**
The LRET phenomenon was widely documented in other cold regions (Figure 4), including
but not limited to the headwater of Yellow River (Yang et al., 2019), and a small headwater
catchment at Cape Bounty Arctic Watershed Observatory, Melville Island, NU in Canada
(Lafrenière and Lamoureux, 2019). For example, at the headwater of Yellow River, on July 8-
9 2014, there was a 21.9mm rainfall event with temperature of 6.5℃, but little 1.5m$^3$/s
runoff; and on July 21-23 2014, there was a 27.4mm rainfall event with temperature of
7.2℃, but only 5.1m$^3$/s runoff. Lafrenière and Lamoureux (2019) found that the undisturbed
frozen soil at Cape Bounty Arctic Watershed Observatory had little runoff generation in the
early thawing season. But after the frozen-soil was disturbed, the runoff response to rainfall
event was much larger. Hence this paired catchment study illustrated that frozen soil played
a key role causing the LRET phenomenon.
## 3.2 Observation2: Discontinuous baseflow recession (DBR)
Baseflow recession provides an important source of information to infer groundwater
characteristic, including its storage properties, subsurface hydraulics, and concentration
times (Brutsaert and Sugita, 2008; Fenicia et al., 2006), which is especially true for basins with
frozen-soil (Ye et al., 2009; Song et al., 2020).
Baseflow analysis is based on the water balance equation (Equation 1), and linear reservoir
assumption (Equation 2). It should be noted that Equation 1 assumes no additional inflows
(recharge or thawing) or outflow (capillary rise or freezing). If a reservoir is linear, this
implies that the reservoir discharge ($Q$) has a linear relationship with its storage ($S$). $K_s$ (days)
is time-constant controlling the speed of recession in the linear reservoir. With a larger $K_s$
value, the reservoir empties slower, and vice-versa. Combining Equation 1 and 2, we can
derive equation 3, illustrating how discharge depends on time ($t$), proportional to the initial
discharge ($Q_0$).
$\frac{dS}{dt} = -Q$  (1)
$Q = S/K_s$  (2)
$Q = Q_0 \cdot e^{-t/Ks}$  (3)
**Baseflow recession analysis results**
In Figure 5, we plot the groundwater recession on semi-logarithmic scale. In the beginning
of freezing season, baseflow presented a clear linear recession, and was able to be fitted by
setting the recession coefficient ($K_s$) as 80 days. However, interestingly, simultaneously with
the LRET phenomenon, we observed a clear discontinuous baseflow recession in the Hulu
catchment (Figure 5). The baseflow bended down, and $K_s$ became 60 days in the end of
recession periods. This DBR phenomenon informed us that the groundwater system was
disturbed. We also noted the spikes during the thawing in Fig. 5, which means groundwater
was suddenly released. To our best knowledge, these discontinuous baseflow recession
(DBR) phenomenon is a new observation for mountainous frozen-soil hydrology.
We also plot the variation of groundwater depth of 4 wells at WW01 site from 2016 to 2019
(Figure 1, 11). The wells of 5m, 10m, and 15m only had liquid water in thawing seasons, and
gradually went dry in recession periods. The water level of the 25m well was decreasing in
the entire frozen seasons, but in a discontinuous way. From the observed groundwater
depth, the groundwater level decreases faster at beginning. And after the groundwater level
dropped below around 17m, the decrease of groundwater level became slower. The turning
point from $K_s$=80d to $K_s$ =60d occurred simultaneously while the groundwater level went
down to 17m (Figure 5, 11). The bending down discontinuous baseflow recession and
slower decrease of groundwater level indicated that there was a disturbance reduced the
groundwater discharge and lead to a slower decrease of groundwater level.
**Learning from paired catchments**
Frozen-soil process is able to disturb the groundwater system, by freezing the liquid
groundwater to reduce groundwater storage, and leading to the DBR. But the DBR could
also be caused by many other reasons, for instance soil evaporation, root tapping, capillary
rise, impermeable layer, and heterogenous hydraulic conductivities in different landscapes
(riparian area and hillslope). Since the DBR phenomenon started from the middle of freezing
period, and lasted until the end of frozen season, in which time the evaporation, root
tapping, and capillary rise were very inactive even totally stopped, which allows us to
exclude these impacts. The impermeable layer and heterogenous hydraulic conductivities
are both able to result in discontinuous recession in small scale.
To further understand the DBR phenomenon, we collected larger scale runoff data in this
region, including the Zhamashike (5526 km$^2$) and Qilian (2924 km$^2$), which are also the sub-
basins of the upper Heihe. The permafrost and seasonal frozen-soil map of the upper Heihe
River basin shows that: the Zhamashike sub-basin has 74% permafrost and 26% seasonal
frozen soil, which is similar to the Hulu catchment; while the Qilian sub-basin has much less
permafrost area (38%), and is mostly covered by seasonal frozen-soil (62%) (Figure 1).
Interestingly, when plotting the hydrographs of these two sub-basins on logarithmic scale,
we can clearly see that in the permafrost dominated Zhamashike sub-basin discontinuous
recessions occurred, with $K_s$=60d in the early and $K_s$ = 20d in in the end of the recession
period. This DBR phenomenon is the same as we found in the Hulu catchment, although the
Zhamashike and Hulu catchment have quite different scales (5526 km$^2$ versus 23.1 km$^2$)
(Chen et al. 2018, Gao et al., 2014). Discontinuous baseflow recession happened almost
every year at the Zhamashike station, and we only highlight the hydrograph in 1974 to
demonstrate this phenomenon in Figure 6. On the other hand, of similar size as the
Zhamashike, the Qilian sub-basin (2924 km$^2$), which is mostly covered by seasonal frozen-
soil, only has one continuous recession, with $K_s$ = 60d. The results from these two paired
catchments provided good reasons to interpret the DBR phenomenon in the Hulu
catchment as the result of frozen-soil.

## 3.3 Perceptual frozen-soil hydrology model of Hulu catchment

We did an expert-driven data analysis of the LRET and DBR observations, including time
series analysis of precipitation-runoff, soil moisture profile, freeze/thaw depth, and paired
catchments comparison. These comprehensive data analyses allowed us to identify that
both the LRET and the DBR were the results of frozen-soil, and motivated the following
perceptual model (Figure 7).
In the freezing season, at local scale, frozen soil occurs from the top soil downwards. But at
catchment scale, the freezing process does not occur in a homogeneous way. Since the
higher elevation is colder than the lower elevation, the freezing process starts from higher
elevation and downwards. At the beginning of the freezing season, although the top soil is
already frozen, the groundwater discharge from the supra-permafrost layer still continues. It
was found that the groundwater recession from the supra-permafrost layer determines the
dominant part of baseflow in permafrost regions (Brutsaert and Sugita, 2008; Ma et al.,
2021). Thus, the baseflow recession contributed by both permafrost and seasonal frozen-
soil areas, during this period, appears not to be influenced by frozen topsoil, and maintains
a linear recession pattern, with $K_s$=80d in the Hulu catchment.
In the frozen season, the hydrological processes in the topsoil are almost completely
blocked, although there is still very small amount of unfrozen liquid water in the frozen soil.
Thus, surface hydrological processes are almost stopped. With the increase of frozen depth,
groundwater in the supra-permafrost layer is frozen gradually from higher to lower
elevation, which gradually but dramatically reduces groundwater discharge. Interestingly,
permafrost and seasonal frozen-soil have different impacts on baseflow recession. In
seasonal frozen-soil region, with shallower frozen depth, the groundwater is still active, and
continues the discharge. But in the permafrost region, the groundwater in the supra-
permafrost layer is largely inactive. This means the recession, in the frozen periods, was only
contributed by the seasonal frozen-soil area, with faster decline of baseflow, and the bend
down of hydrograph, with $K_s$ decreased to 60 days in Hulu catchment.
During the early thawing season, after a long groundwater recession, the groundwater level
is deep; and due to soil evaporation in winter, soil is dry and deficit of moisture.
Observations show that with the progress of thawing, soil temperature and soil moisture
increase from top to bottom, and the thawing front deepens downward. Rainfall firstly
infiltrates to saturate the moisture deficit without runoff generation. Moreover, the existence
of the impermeable deeper frozen-soil layer leads to vertical disconnection between soil
water and groundwater. Although there is probably saturated water above the frozen soil
(Figure 3 and 7), the poor vertical connectivity largely hinders soil water percolation.
Without recharge to groundwater, there is limited runoff generation during the early
thawing.
At the end of the thawing season, as soon as the thaw depth reaches its maximum or
frozen-soil no longer exists, the frozen groundwater is released. The spikes in the observed
hydrograph are likely due to different parts of the catchment reaching breakthrough, which
happens first in the lower elevations and then gradually moves upward to higher landscape
elements. This would trigger a sequence of sudden groundwater release and the spikes.
Snowfall is probably another influencing factor, causing the LRET and the spikes
phenomenon by storing precipitation as snow cover to reduce runoff, and releasing melting
water in a short time. This hypothesis needs to be tested by including snow accumulation
and melting processes in the hydrological model.
Once the soil is completely thawed at the end of the melting season, the rainfall-runoff
process returns to normal, and free of influence by the frozen-soil. Hence in the Hulu
catchment, frozen soil mainly impacts on streamflow during the freezing, frozen and
thawing periods.
Based on this perceptual model, we developed the conceptual framework of the frozen-soil
hydrological model, which needs to at least to consider the following elements: 1) we need
a semi-distributed modeling framework; 2) distributed forcing should be considered. 3)
different landscapes should be included; 4) topography should be involved, particularly
elevation; 5) we need to consider soil freeze/thaw processes; 6) snowfall and melting should
be included; 7) glacier melting should be included; 8) last but not least, the normal rainfall-
runoff processes are still important, because most runoff happen in the warm season, which
functions the same as in temperate climate regions.

# 382 4 A semi-distributed conceptual frozen-soil hydrology model

The perceptual model requires a quantitative conceptual model to test, revise, polish, verify,
or even reject its hypotheses. Since the Hulu catchment has heterogenous landscapes, with
a large elevation gradient, diverse land cover, and complex freeze/thaw process, we

developed a semi-distributed frozen-soil hydrological model, i.e. FLEX-Topo-FS, based on the landscape-based hydrological modeling framework, i.e. FLEX-Topo. Numerous processes were involved in FLEX-Topo, including the distributed meteorological forcing, landscape heterogeneity, and snow and glacier melting. And in FLEX-Topo-FS, we explicitly considered the impacts of frozen-soil on soil water percolation and groundwater frozen processes in the supra-permafrost layer. The details of both the FLEX-Topo model (without frozen soil) and FLEX-Topo-FS model (with frozen soil) are described in below.

## 4.1 FLEX-Topo model (without frozen-soil)

### 4.1.1 Catchment discretization and meteorological forcing interpolation

The FLEX-Topo model classified the entire Hulu catchment into four landscapes, i.e. glaciers, alpine desert, vegetation hillslope, and riparian zone. The Hulu catchment (from 2960m to 4820m) was classified into 37 elevation bands, with 50m interval (Figure 2). Combined 4 landscapes and 37 elevation bands, we had 37×4=148 hydrological response units (HRUs). The structure of FLEX-Topo model consisted of four parallel components, representing the distinct hydrological function of different landscape elements (Savenije, 2010; Gao et al., 2014; Gharari et al., 2014; Gao et al., 2016). And the corresponding discharge of all elements was subsequently aggregated to obtain the simulated runoff.

We interpolated the precipitation ($P$) and temperature ($T$) based on elevation bands from in-situ observation (2980m) to each elevation band. The precipitation increasing rate was set as 4.2%/100m, and temperature lapse rate as -0.68°C/100m, based on field measurements (Han et al., 2013). Snowfall ($P_s$) or rainfall ($P_l$) was separated by air temperature, with the threshold temperature as 0°C (Gao et al., 2020).

### 4.1.2 Model description and configuration

FLEX-Topo is a semi-distributed conceptual bucket model (Savenije, 2010; Gao et al., 2014), with three modules, i.e. the snow and glacier module, the rainfall-runoff module, and the groundwater module. The water balance and constitutive equations can be found in Table 1. The model parameters and their prior ranges for calibration are listed in Table 2.

**Snow and glacier module**

The temperature-index method was employed to simulate snow and glacier melting (Gao et al., 2020; He et al., 2021). We used a snow reservoir ($S_w$) to account for the snow accumulating, melting ($M_w$) and water balance (Equation 4). The snow degree-day factor ($F_{dd}$) needs to be calibrated. For glaciers, we assumed its area was constant in our simulation (from 2011 to 2014). Glacier melting ($M_g$) was also calculated by the temperature-index method (Equation 7), but with different degree-day factor. With the same air temperature, glacier has less albedo than snow cover, thus with larger amount of melting. Glacier degree-factor was obtained by multiplying snow degree-day factor ($F_{dd}$) with a correct factor $C_g$ (Equation 8) (Gao et al., 2020).

**423  Rainfall-runoff module**

There are two reservoirs to simulate rainfall-runoff process, including the root zone
reservoir ($S_u$) (Equation 10) and fast response reservoir ($S_f$) (Equation 17). To account of the
different rainfall-runoff processes in different landscapes and simultaneously avoid over
parameterization, we kept the same model structure for vegetation hillslope, riparian and
alpine desert (Equation 11, 12), but gave different root zone storage capacity ($S_{umax}$) values,
i.e. $S_{umax\_R}$ for riparian, $S_{umax\_D}$ for cold desert, and $S_{umax\_V}$ for hillslope vegetation. For vegetation
hillslope, a larger prior range was constrained for the root zone storage capacity ($S_{umax\_V}$),
which means more water is required to fill in its storage capacity to meet its water deficit,
which is evidenced by previous studies in this region (Gao et al., 2014). For alpine desert,
due to its sparse vegetation cover, we constrained a shallower root zone storage capacity
($S_{umax\_D}$). For the riparian area, due to its location where is prone to be saturated, we also
constrained a shallower root zone storage capacity ($S_{umax\_R}$). The initial states (beginning of
2011) of the reservoirs were obtained from the end values (end of 2014) of the simulation,
which is a normal procedure in modeling practice.
Other parameters in rainfall-runoff module (Equation 11-18, Table 1) include the threshold
value controlling evaporation ($C_e$), the shape parameter of the root zone reservoir ($\beta$), the
splitter ($D$) separating the generated runoff ($R_u$) from the root zone reservoir ($S_u$) to the fast
response reservoir, the recession parameter of faster reservoir ($K_f$), and the lag time from
rainfall event to peak flow ($T_{lagF}$). We set $D$ as 0.2 from the isotope study (Ma et al., 2021).
And other parameters to be calibrated, with prior ranges (Table 2) based on previous
studies (Gao et al., 2014; Gao et al., 2020).

**445  Groundwater module**

The baseflow ($Q_s$) is generated from groundwater recession. The groundwater was
simulated by a linear reservoir ($S_s$) described in Section 3.2, and Equation 19. We set the
prior range for recession coefficient of baseflow reservoir ($K_s$) as (10-100 d). To estimate the
impacts of frozen groundwater on hydrological processes, we set the groundwater in
different landscapes as parallel. But we analyzed groundwater level as an integrated system,
because groundwater system is connected and this affects the groundwater level. Since the
sub-permafrost groundwater is even deeper than 20m in the Hulu catchment, and almost
disconnected to streamflow, thus we only model the supra-permafrost groundwater.
$$Q_s = S_s / K_s \text{ (19)}$$

**455  4.2 FLEX-Topo-FS model (with frozen soil)**

**456  4.2.1 Modeling the soil freeze/thaw processes**

FLEX-Topo-FS model employed the Stefan equation (Equation 20), to provide an
approximate solution to estimate freeze/thaw depth (Figure 9). The Stefan equation is a
temperature-index based freeze-thaw algorithm, which assumes the sensible heat is
negligible in soil freeze/thaw simulation (Xie and Gough, 2013). The form of Stefan equation
is written as:
$\varepsilon = \left(\frac{2 \cdot 86400 \cdot k \cdot F}{Q_L}\right)^{0.5} = \left(\frac{2 \cdot 86400 \cdot k \cdot F}{L \cdot \omega \cdot \rho}\right)^{0.5}$ (20)
where $\varepsilon$ is the freeze/thaw depth; $k$ is the thermal conductivity (W/(m·K)) of the soil; $F$ is the
surface freeze/thaw index. Freeze index (℃ degree-days) is the accumulated negative
ground temperature, while freezing; thaw index (℃ degree-days) is accumulated positive
ground temperature, while thawing. $Q_L$ is the volumetric latent heat of soil, in J/m$^3$; and
$Q_L = L \cdot \omega \cdot \rho$ where $L$ is the latent heat of fusion of ice (3.35·10$^5$ J/kg); $\omega$ is the water
content, as a decimal fraction of the dry soil weight; and $\rho$ is the bulk density of the soil
(kg/m$^3$).
We set the thermal conductivity as $k$=2 W/(m·K), the water content as a decimal fraction of
the dry soil weight $\omega$ =0.12, and bulk density of the soil $\rho$=1000 kg/m$^3$ (Zhang et al.,
2019). Since the Stefan equation requires ground surface temperature, which is difficult to
measure and often lack of data, we used a multiplier to translate the air temperature to
ground temperature. The multiplier during freezing was set as 0.6, and during thawing we
assumed the ground surface temperature was the same as air temperature (Gisnås et al.,
476  2016).

In this model, we did not consider the impacts of snow cover on soil freeze/thaw, because
the snow effects were compiled in the Hulu catchment. Firstly, because precipitation in the
Hulu catchment mostly happens in summer as rainfall, and snow depth in the Hulu
catchment was less than 10mm in most area and time. Secondly, the snow cover has
contrary effects on ground temperature. Snow cover as an isolation layer increased ground
temperature in winter. But simultaneously snow cover also increased albedo, which
decreased net radiation, and decreased ground temperature. To avoid over
parameterization, we did not consider snow effect in the Stefan equation.
In this study, the Stefan equation was driven by distributed air temperature, which allowed
us to simulate the distributed soil freeze/thaw processes. With the distributed soil freeze
index and thaw index, we can also estimate the lower limit of permafrost, of which elevation
the freeze index equals to the thaw index in mountainous regions. Field survey on the lower
limit of permafrost (Wang et al., 2016) can provide another strong confirmation to our
simulated soil freeze/thaw process, except for the spot-scale freeze/thaw depth.

### 4.2.2 Modeling the impacts of frozen-soil on hydrology

The distributed freeze/thaw status calculated by FLEX-Topo-FS model allowed us to
simulate the impacts of frozen-soil on soil and groundwater systems, their connectivity, and
eventually catchment runoff.
In freezing and frozen seasons, precipitation was in the phase of snowfall, and topsoil was
frozen, thus without surface runoff. During this period, runoff is only contributed from the
groundwater discharge of the supra-permafrost layer ($Q_s$). There is no runoff generation ($R_u$)
from the root zone reservoir to the response routine ($S_s$ and $S_f$) in this period. In the
conceptual model, we set $R_u = 0$ (Equation 11). In freezing season, when frozen depth was
less than 3m (the depth of active layer in this region), the entire groundwater in the supra-
permafrost layer were still connected, and could be simulated by linear groundwater
reservoir ($S_s$). Once the frozen depth of certain elevation zone is larger than 3m, the
groundwater in that elevation zone was frozen ($F_s$). In the FLEX-Topo-FS model, we reduced
the groundwater storage ($S_s$) to 10% of its total storage, to simulate its frozen status
(Equation 21, 22). This amount of frozen water ($F_s$, 90% of groundwater storage when frozen,
marked as $S_s(\tilde{t})$ ) was held in the groundwater system as frozen-soil (Equation 22), but not
disappeared. We set 90% frozen, rather than 100%, because there is still unfrozen liquid
water in frozen-soil (Romanovsky and Osterkamp, 2000). Groundwater discharge was
controlled by the frozen status, which was frozen from high elevations to lower elevations.
This process is progressively stopping the function of a series of cascade groundwater
buckets, resulting in the discontinuous recession. Simultaneously, the decrease of discharge
($Q_s$) slowed down the decrease of groundwater level ($S_s$). This conceptual model allowed us
to simulate the bend down of baseflow recession and slower decreasing of groundwater
level.
$$\frac{\mathrm{d}S_s}{\mathrm{d}t} = R_s - Q_s - F_s \quad (21)$$
$$F_s = \begin{cases} 0.9 \cdot S_s(\tilde{t}); & \text{once } freeze\ depth \geq 3m \\ -0.9 \cdot S_s(\tilde{t}); & \text{once } thaw\ depth \text{ reach to yearly max} \\ & \text{or } thaw\ depth\ \geq\ freeze\ depth \end{cases} \quad (22)$$
In thawing seasons, the freeze/thaw condition in the lowest elevation zone plays a key role,
controlling the hydraulic connectivity between soil and groundwater systems. In the
conceptual model, if freeze depth calculated by Stefan equation is larger than thaw depth,
this means the frozen layer still exists, which obstructs the soil and groundwater connection.
In the conceptual model, we kept as the runoff generation $R_u = 0$ (Equation 11). Since there
is no percolation from soil to groundwater, and root zone soil moisture ($S_u$) is accumulating,
even ponding in some local depressions (Figure 7). The only outflow of the root zone is
evaporation in this period. This conceptual model allowed us to reproduce the LRET
observation. For groundwater reservoir, once the thaw depth goes to its yearly maximum (in
permafrost area) or thaw depth > freeze depth (in seasonal frozen-soil area), the frozen
water (90% of groundwater storage when frozen, $S_s(\tilde{t})$ ) was released to the groundwater
again (Equation 22). The sudden release of frozen groundwater causes the spikes in
hydrograph, which could happen in either thawing seasons or complete thaw seasons
depending on its elevation. But in either way, the water balance calculation is one-hundred-
percent closed.
Complete thaw in the lowest elevation marked the end of thawing season, and the start of
complete thaw season. In the complete thaw season, soil water and groundwater are
connected, and runoff generation ($R_u$) returns to normal circumstances, which can be
simulated by the FLEX-Topo model without frozen-soil.

## 4.3 Model uncertainty analysis and evaluation metrics

The Kling-Gupta efficiency (Gupta et al., 2009; KGE) was used as the performance metric in
model calibration:

$$KGE = 1 - \sqrt{(r-1)^2 + (\alpha-1)^2 + (\beta-1)^2} \tag{23}$$

Where $r$ is the linear correlation coefficient between simulation and observation; $\alpha$ ($\alpha =$
$\sigma_m/\sigma_o$) is a measure of relative variability in the simulated and observed values, where $\sigma_m$ is
the standard deviation of simulated variables, and $\sigma_o$ is the standard deviation of observed
variables; $\beta$ is the ratio between the average value of simulated and observed variables.
We applied the Generalized Likelihood Uncertainty Estimation framework (GLUE, Beven and
Binley, 1992) to estimate model parameter uncertainty. Sampling the parameter space with
20, 000 parameter sets, and select the top 1% parameter as behavioral parameter sets.
For a comprehensive assessment of model performance in validation, the behavioral model
runs were evaluated using multiple criteria, including KGE, KGL (the KGE of logarithms flow,
and more sensitive to baseflow), Nash-Sutcliffe Efficiency (NSE) (Nash and Sutcliffe, 1970)
(Equation 24), coefficient of determination ($R^2$) and root mean square error (RMSE).
$NSE = 1 - \frac{\sum_{t=1}^{n}(Q_o - Q_m)^2}{\sum_{t=1}^{n}(Q_o - \overline{Q_o})^2}$ (24)
Where $Q_o$ is observed runoff, $\overline{Q_o}$ is the observed average runoff, and $Q_m$ is modeled runoff.
The model was calibrated in the period 2011-2012, and uses KGE as objective function. The
second half time series (2013-2014) were used to quantify the model performance in
streamflow split-sample validation, with multi-criteria including KGE, KGL, NSE, $R^2$, and
RMSE. The KGE, KGL, NSE and $R^2$ are all less than 1, and their valuation closer to 1 indicates
better model performance. While the less value of RMSE indicates less error and better
performance.

## 5 Testing the realism of FLEX-Topo-FS model

### 5.1 FLEX-Topo model results and its discrepancy

Figure 10 shows that the FLEX-Topo model can somehow reproduce the observed
hydrography in most periods, except for the LRET and DBR events. In calibration, the KGE
was 0.78. And in validation, the KGE =0.58, KGL =0.36, NSE =0.41, $R^2$= 0.82, and RMSE =
0.95mm/d. While taking account the impacts of landscape heterogeneity, the FLEX-Topo
model can to some extent simulate the LRET phenomenon. The vegetated hillslope in
relatively lower elevation has larger unsaturated storage capacity, with larger soil moisture
deficit in the beginning of melting season, and capable to hold more rainfall with initial dry
soil. Moreover, FLEX-Topo model has took snow accumulation and melting into account,
which also reduced the runoff generation during the LRET periods. However, there was still
large overestimation in the early thawing season.
Additionally, the simulated hydrography on logarithm scale clearly shows that the baseflow
is the result of a linear reservoir (Figure 10). The linear reservoir model can mimic recession
quite well in the beginning of baseflow recession, but the model discrepancy becomes
larger in the middle to the end of frozen season. Hence, FLEX-Topo model is not able to
simulate the discontinuous recession. The model discrepancy indicates that without
considering frozen-soil, FLEX-Topo cannot well reproduce the observed LRET and DBR
observations, although explicitly considered landscape heterogeneity, snow and glacier
processes.

## 579 5.2 FLEX-Topo-FS model results

### 580 5.2.1 Freeze/thaw simulation by FLEX-Topo-FS model

Figure 9 demonstrates that the Stefan equation was capable to reproduce the freeze/thaw
process. This verified the success of the freeze/thaw parameterization and the parameter
sets. Also, the simulated lower limit of permafrost is 3716m, which is largely close to field
survey in the upper Heihe River basin, around 3650 – 3700 m (Wang et al., 2016), and the
expert-based estimation of 3650 m of Hulu catchment. Both the well reproduced
freeze/thaw variation in spot scale, and the lower limit of permafrost in catchment scale,
gave us strong confidence to the simulation of soil freeze/thaw processes.

### 588 5.2.2 Runoff simulation by FLEX-Topo-FS model

While considering the impacts of frozen-soil, the FLEX-Topo-FS model, compared with
FLEX-Topo, dramatically improved the model performance. Figure 10 showed the simulated
hydrograph by FLEX-Topo-FS on both normal and log scales. Both the LRET and the DBR
observations were almost perfectly reproduced by the FLEX-Topo-FS model. The KGE of
FLEX-Topo-FS in calibration was 0.78, which was the same as FLEX-Topo. But in validation,
the performance was significantly improved, the KGE improved from 0.58 to 0.66, KGL was
from 0.36 to 0.72, NSE was from 0.41 to 0.60, $R^2$ from 0.82 to 0.83, and RMSE was reduced
from 0.95mm/d to 0.79mm/d. All the model evaluation criteria were improved. The most
significant improvement was the baseflow simulation, and KGL was increased from 0.36 to
0.72. We also noted that the FLEX-Topo-FS model reproduced the spikes during the
thawing in Figure 10. This further confirms our conceptual model of a sequence of thawing
breakthroughs, which trigger the sudden release of groundwater starting at lower elevations
and progressing to higher landscape elements.
5.2.3 Modeling groundwater trends
To further verify the FLEX-Topo-FS model, we averaged the simulated groundwater storage
($S_s$) of all HRUs, and compared with the observed groundwater depth on log scale (Figure
11). We included the frozen groundwater in the total groundwater storage ($S_s$), because the
liquid groundwater is in connection with the frozen groundwater and this affects the
groundwater level variation. Figure 11 clearly demonstrated that the simulated groundwater
storage decreased slower, and the time scale of recession was increased. The trends of
simulated groundwater storage and observed groundwater level, which are not the same,
but similar physical meaning describing groundwater dynamic, correspond surprisingly well.
This is particularly encouraging, given that the periods of simulation (2011-2014) and
observation (2016-2019) were not overlaid, and a point observation may not be
straightforwardly representative for the entire basin.
The success to reproduce groundwater level trends is another strong confirmation for the
FLEX-Topo-FS model. All the successes of FLEX-Topo-FS model to reproduce spot scale
freeze/thaw depth variation, lower limit of permafrost, LRET and DBR events, and
groundwater level trends, gave us strong confidence to the realism of our qualitative
perceptual model and quantitative conceptual model.
# 6 Discussion
## 6.1 Diagnosing the impacts of frozen-soil on complex mountainous hydrology
### 6.1.1 Understanding complex frozen-soil hydrology by hydrography analysis
Frozen-soil happens underneath, with frustrating spatial-temporal heterogeneities, and
difficulty to measure. Although there are spot and hillslope measurements, its impact on
catchment hydrology is still hard to explore. Hydrography, easily and widely observed and
globally accessible, can be regarded as the by-product of the entire catchment hydrological
system (Gao, 2015). Hydrography as an integrated signal provides us a vital source of
information, reflecting how the complex hydrological system works, i.e. transforming
precipitation into runoff. Hence, hydrography itself is a valuable source of data to
understand catchment frozen-soil hydrology.
Especially the baseflow embodies the influence of basin characteristics including the
geology, soils, morphology, vegetation, and frozen-soil (Blume et al., 2007; Ye et al., 2009).
Hence, the quantitative description of baseflow is a valuable tool for understanding how the
groundwater system behaves (McNamara et al., 1998). Baseflow recession was used to
identify the impacts of climate change on permafrost hydrology. In previous studies,
Slaughter and Kane (1976) found that basins with permafrost have higher peak flows and
lower baseflows. The baseflow, representing groundwater recession, provides important
information about the storage capacity and recession characteristics of the active layer in
permafrost regions (Brutsaert, and Hiyama, 2012).
Moreover, hydrological system has tremendous influencing factors. The hydrograph of
paired catchments provides a good reference, as a controlled experiment, to isolate one
influencing factor from the others. Nested catchments helped us to acknowledge the
importance of region-specific knowledge, which is often the key to interpret the
unexplained variability of large sample studies (Fenicia and McDonnell, 2022). In this study,
the pair catchment method helped us to confirm the impacts of frozen-soil on LRET and
DBR observations.
Additionally, by analyzing the nested sub-basins of the Lena River in Siberia, Ye et al. (2008)
used the peak flow/baseflow ratio to quantify the impact of permafrost coverage on
hydrograph regime in Lena River basin, and found that frozen-soil only affects discharge
regime over high permafrost regions (greater than 60%), and no significant affect over the
low permafrost (less than 40%) regions. In this study, we reconfirmed this statement. The
permafrost area proportions of the Hulu catchment and Zhamashike sub-basin are 64% and
74%, with significant effects on discharge, while 38% of the Qilian sub-basin is covered by
permafrost, with no significant effects on discharge regime.
By paired catchments comparison, interestingly, the $K_s$ in Zhamashike and Qilian in the early
recession period are both 60d, which is exactly within the standard value of 45±15 days
derived in earlier studies for basins ranging in size between 1,000 and 100,000 km$^2$
(Brutsaert and Sugita, 2008; Brutsaert and Hiyama, 2012), which was likely the results of
catchment self-similarity and co-evolution. But we also noticed that the $K_s$ in the small Hulu
catchment ($K_s$ = 80d and 60d) is quite larger than the Zhamashike ($K_s$ = 60d and 20d) and
Qilian ($K_s$ = 60d). This could be rooted in different scale and drainage density (Brutsaert and
Hiyama, 2012). The Hulu catchment is located in the headwater with less drainage density,
hence less contact area between hillslope and river channel, slower baseflow recession, and
larger $K_s$ value. We argue that even without the impacts from frozen-soil, it is difficult to
give accurate $K_s$ estimation in small catchments (less than 1,000 km$^2$) in moderate climates. It
is more substantial difficult to estimate the discontinuous $K_s$ in different periods in frozen-
soil catchments without calibration. Thus, estimating the value of recession coefficient ($K_s$) in
different catchments and periods, especially for small catchments and in cold regions, is still
an intriguing scientific question for hydrologists.

## 6.1.2 Understanding complex frozen-soil hydrology by multi-source observations

Observation is still a bottleneck in complex mountainous cold regions. Traditionally,
fragmented observations are only for specific variables, like puzzles. In this study, we
collected multi-source data, including soil moisture, groundwater level, topography,
geology survey, isotope, soil temperature, freeze/thaw depth, permafrost and seasonal
frozen-soil map, and hydrograph in paired catchments. Multi-source data analysis provides
multi-dimensional perspective to investigate frozen-soil hydrology. We argue that on one
hand, multi-source observations helped us to deliver the perceptual and conceptual
models. And on the other hand, perceptual and conceptual models bridge the gap between
experimentalists and modelers (Seibert and McDonnell, 2002), allowing us put fragmented
observations together, and understand the hydrological system in an integrated and
qualitative way.
Data gap is a common issue in mountainous hydrology studies. For example, in this study,
runoff data has a gap period in the end of thawing season in 2013, due to flooding and
equipment malfunction. Soil moisture and groundwater level data had large gap, which
cannot be used for continuous modeling. Luckily, the meteorological data, which is
important forcing data to run the models, was continuous without any gap. With sufficient
meteorological forcing data, we successfully run the hydrological models from 2011 to
2014. The runoff data gap in the end of thawing season in 2013 merely influenced model
validation. While evaluating models, we did not involve the data gap period. Hence, the
data gap does not have any impact on the consolidation of the conclusions.
Although soil moisture had large gap, fortunately there were some observations during the
LRET periods, which were sufficient to distinguish the impacts of frozen soil on soil moisture
profile. Additionally, although the groundwater level observation (2016-2019) was not
overlaid with other hydrometeorological measurements (2011-2014), its repeating seasonal
pattern allowed us to qualitatively understand how groundwater system behaves.
Groundwater fluctuation in natural catchments has strong periodicity, which can be
observed in Figure 11. Groundwater variation does not show significant difference among
different years, which is especially true for the 25m well. Due to the extreme difficulty of
continuous observation in this region, there was no groundwater measurement in 2011-
2014. But due to the strong repeated temporal variation of groundwater level, we have
good reason to believe the trends in 2016-2019 also happened in 2011-2014. Moreover,
this is a qualitative comparison, rather than a quantitative one, which we do not think has
any impact on our conclusions.
In general, we argue that data gap always exists. In another words, we can never have
sufficient data. The only thing we can do is using the accessible data to understand
processes. Although perfect data does not exist, with more multi-source and better data
quality, the more accurate understanding we can achieve. This needs the close collaboration
among multi-disciplinary researchers, including but not limited to hydrologists,
meteorologists, ecologists, geocryologists, geologists, and engineers.

### 6.1.3 Understanding complex frozen-soil hydrology by model discrepancy


By a simple water balance inspection, we found that the total annual runoff of Hulu
catchment was 499mm/a, which is even larger than the observed annual precipitation
433mm/a. This means that without considering distributed meteorological forcing, the
runoff coefficient is larger than 1, and the water balance cannot be closed. This result is also
in line with previous studies, showing that precipitation in mountainous areas is largely
underestimated (Immerzeel et al., 2015; Chen et al., 2018; Zhang et al., 2018b).
Although the semi-distributed FLEX-Topo has considered tremendous processes, including
rainfall-runoff processes, distributed forcing, landscape heterogeneity, topography, snow
and glacier melting, there was still model discrepancy to reproduce the LRET and DBR
observations. This means there must be some processes missed in the model. After our
expert-driven data analysis, we attributed the model discrepancy to soil freeze/thaw
processes.
Model fitness is the goal which all modelers are pursuing. But we argue that in many cases,
model discrepancy can tell us more interesting things than perfect fitting. In this study, we
used the FLEX-Topo model, without frozen-soil, as a diagnosing tool to understand the
possible impacts of frozen-soil on the complex mountainous hydrology. Using tailor-made
hydrological model and integrated observations as diagnostic tools is a promising approach
to step-wisely understand the complex mountainous hydrology.

## 6.2 Modeling frozen-soil hydrology: top-down VS bottom-up


Top-down and bottom-up are two philosophies for model development (Sivalpalan et al.,
2003). The bottom-up approach attempts to model catchment scale response based on the
prior knowledge learned in small scale. Bottom-up approach is commonly used in frozen-
soil hydrological modeling, because this is straightforward. And for modeling, it is a
common practice to experiment with/without a certain process, and claim its impacts on
runoff. But the bottom-up modeling largely missed the key step to diagnose the impacts of
small-scale processes on catchment response. Lack of process understanding usually leads
modeling studies to data pre- and pro-processing and extensive parameter calibration with
the risk of equifinality and model malfunction.
By our expert-driven top-down modeling approach, we firstly tried to understand the
hydrological processes at work, using multi-source data and analysing model discrepancy.
We then translated our understanding to perceptual and conceptual models. The top-down
method is an appealing way to identify the key influencing factors, rather than being lost in
endless details and heterogeneities. Such informed analysis of the data helps to bring
experimentalist insights into the initiation of the conceptual model construction.

## 6.3 Warming impacts on frozen-soil hydrology


To quantify the impacts of warming on frozen-soil hydrology, we arbitrarily set the air
temperature increased by 2°C. The FLEX-Topo-FS simulation illustrated that complete thaw
date became 16-19 days earlier, and the lower limit of permafrost increased by 294 m, from
3716m to 4010m. For runoff simulation, the DBR phenomenon became less obvious (Figure
12). This is because two-degree warming results in permafrost degradation, which means
most permafrost is degraded to seasonal frozen-soil. Since the DBR was caused by the
different groundwater discharge behaviors in permafrost and seasonal frozen-soil areas.
Specially, the first recession period was contributed by the groundwater discharge from
both permafrost and seasonal frozen-soil areas, and the second recession period was only
from the seasonal frozen-soil area. The permafrost degradation turns most permafrost into
seasonal frozen-soil, and makes groundwater discharge nearly only from the seasonal
frozen-soil region, and leads to more continuous baseflow recession. Eventually, warming
leads to the increase of both the baseflow and runoff in early thawing seasons. The warming
effect on baseflow was already widely observed in Arctic and mountainous permafrost rivers
(Ye et al., 2009; Brutsaert et al., 2012; Niu et al., 2010; Song et al., 2020). Hence, these wide
observations could be another verification for the FLEX-Topo-FS model realism. For
implications in water resource management, the results indicate that frozen soil degradation
caused by climate change may largely alter streamflow regime, especially for the thirsty
spring and early summer, in vast cold QTP. It is also worthwhile to be noted that this is
primary prediction. We used 2 degrees warmer more like using a sensitive analysis to
illustrate how warming will impact on baseflow, to further verify the capability and
robustness of the model itself. In future studies, we need more detailed modelling studies to
use the state-of-the-art climate prediction and downscaling methodologies, to assess the
frozen-soil change and hydrology variations.
## 6.4 Implications for other cold regions
We believe that the FLEX-Topo-FS model has great potential to be applied in other cold
regions. There are mainly three reasons.
Firstly, our study site, the Hulu catchment, although small (23.1 km$^2$), has a large elevation
gradient (from 2960 m to 4820 m), diverse landscapes (hillslope vegetation, riparian area,
alpine desert, and glaciers), snowfall and snowmelt, and both permafrost and seasonal
frozen-soil. Our newly developed model explicitly considered all these spatial and temporal
heterogeneities, and eventually achieved excellent performance. With such a comprehensive
modeling toolkit, the model has potential to be upscaled or transfer to other cold regions.
Secondly, we obtained the perceptual model from not only the observations and our expert
knowledge at the Hulu catchment itself, but also widely considered the impact of frozen-
soil on hydrological processes in other catchments, including the Zhamashike and Qilian
(two nested sub-catchments of the upper Heihe), the headwater of Yellow River, and the
Cape Bounty Arctic Watershed Observatory in Canada. Thus, we developed the model for
the Hulu catchment in the context of larger scale observations.
Thirdly, the realism of the conceptual model was confirmed not only by streamflow
measurement, but also by multi-source and multi-scale observations, particularly the
freezing and thawing front in the soil, the lower limit of permafrost, and the trends in
groundwater level variation.
Although our new model generally has great potential to be used in other cold regions, we
should be cautious to arbitrarily use the model without any prior understanding of the
modeling system. Since frozen-soil is merely one influential factor for cold region
hydrology, there are other factors having notable impacts, which are intertwined with
frozen-soil. This relates especially to the geology condition, which can have considerable
impact on frozen-soil, but has large spatial heterogeneity, and where it is difficult to take
measurements. For the model empirical parameters, most of them are related to the freeze-
thaw processes related to Stefan equation, which have clear physical meanings, and
confirmed by previous studies with a good spatial distribution over the entire QTP (e.g. Zou
et al., 2017; Ran et al., 2022). Due to the extreme complexity of soil and geology in
mountainous catchment, we still need to recalibrate their values while modeling other
basins on the QTP. Hence, before upscaling to other cold regions, we recommend to follow
a stringent modeling procedure, i.e expert-driven data analysis → qualitative perceptual
model → quantitative conceptual model → testing of model realism.

# 7 Conclusions

Our knowledge on frozen-soil hydrology is still incomplete, which is particularly true for
complex mountainous catchment on the QTP. In the past decades, we have collected
numerous heterogeneities and complexities in frozen-soil regions, but most of these
observations are still neither well integrated into hydrological models, nor used to constrain
model structure or parameterization in catchment-scale studies. More importantly, we still
largely lack quantitative knowledge on which variables play more dominant roles at certain
spatial-temporal scales, and should be included in models with priority.
By conducting this frozen-soil hydrological modeling study for the complex mountainous
Hulu catchment, we reached the following conclusions: 1) we observed two new
phenomena in the frozen-soil catchment, i.e. the low runoff in early thawing seasons (LRET)
and discontinuous baseflow recession (DBR), which are widespread but not yet reported; 2)
without considering the frozen-soil, the FLEX-Topo model was not able to reproduce LRET
and DBR observations; 3) considering frozen-soil impacts on soil-groundwater connectivity,
and groundwater recession, the FLEX-Topo-FS model successfully reproduced the LRET and
DBR events. The FLEX-Topo-FS results were also verified by observed freeze/thaw depth
variation, groundwater level, and lower limit of permafrost. We believe this study is able to
give us new insights into further implications to understand the impact of frozen soil on
hydrology, projecting the impacts of climate change on water resources in vast cold regions,
which is one of the 23 major unsolved scientific problems in hydrology community.

**ACKNOWLEDGMENTS**
We thank two anonymous reviewers for their constructive comments and suggestions,
which helped us to improve the paper. This study was supported by the National Natural
Science Foundation of China (Grant Nos. 42122002, 42071081, 42171125, and 41971041).

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

**Tables**
Table 1. The water balance and constitutive equations used in FLEX-Topo-FS model. In Equation
10, 11, and 12, the $S_u$ and $S_{umax}$ represent root zone reservoirs and their storage capacities in
different landscapes, including vegetation hillslope ($S_{umax\_V}$), alpine desert ($S_{umax\_D}$) and riparian
($S_{umax\_R}$).

| reservoirs | Water balance equations | Constitutive equations |
|---|---|---|
| Snow reservoir | $\dfrac{\mathrm{d}S_w}{\mathrm{d}t} = P - M_w$ (4) | $P_s = \begin{cases} 0; & T > 0 \\ P; & T \le 0 \end{cases}$ (5) $\quad$ $M_w = \begin{cases} F_{dd} \cdot T; & T > 0 \\ 0; & T \le 0 \end{cases}$ (6) |
| Glacier reservoir | $\dfrac{\mathrm{d}S_g}{\mathrm{d}t} = P_l + M_g - Q_g$ (7) | $M_g = \begin{cases} F_{dd} \cdot T \cdot C_g; & S_w = 0 \text{ and } T > 0 \\ 0; & S_w > 0 \text{ or } T \le 0 \end{cases}$ (8) $\quad$ $Q_g = S_g / K_f$ (9) |
| Root zone reservoir | $\dfrac{\mathrm{d}S_u}{\mathrm{d}t} = P_l + M_w - E_a - R_u$ (10) | $R_u = (P_l + M_w) \cdot (1 - (1 - \dfrac{S_u}{S_{umax}})^{\beta})$ (11) $\quad$ $E_a = E_p \cdot (\dfrac{S_u}{C_e \cdot S_{u\max}})$ (12) |
| Splitter and lag function | | $R_f = R_u D$ (13); $\quad R_s = R_u(1 - D)$ (14) $\quad$ $R_{fl}(t) = \sum_{i=1}^{T_{lagf}} c_f(i) \cdot R_f(t - i + 1)$ (15) $\quad$ $c_f(i) = i / \sum_{u=1}^{T_{lagf}} u$ (16) |
| Fast reservoir | $\dfrac{\mathrm{d}S_f}{\mathrm{d}t} = R_f - Q_f$ (17) | $Q_f = S_f / K_f$ (18) |

Table 2. The parameters of the FLEX-Topo-FS model, and their prior ranges for calibration.

| Parameter | Explanation | Prior range for calibration |
|---|---|---|
| $F_{dd}(\text{mm} \cdot {}^\circ\text{C}^{-1} \cdot \text{d}^{-1})$ | snow degree-day factor | (1-5) |
| $C_g$ (-) | Glacier degree-factor multiplier | (1-3) |
| $S_{umax\_V}$ (mm) | Root zone storage capacity for vegetation hillslope | (50, 200) |
| $S_{umax\_D}$ (mm) | Root zone storage capacity for alpine desert | (10, 100) |
| $S_{umax\_R}$ (mm) | Root zone storage capacity for riparian | (10, 100) |
| $\beta$ (-) | The shape of the storage capacity curve | (0, 1) |
| $C_e$ (-) | Soil moisture threshold for reduction of evaporation | (0.1, 1) |
| $D$ (-) | Splitter to fast and slow response reservoirs | 0.2 |
| $T_{lagF}$ (days) | Lag time from rainfall to peak flow | (0.8, 3) |
| $K_f$ (days) | fast recession coefficient | (1, 10) |
| $K_s$ (days) | baseflow recession coefficient | (10, 100) |
| $k$(W/(m·K)) | thermal conductivity | 2 |
| $\omega$ (-) | water content, as a decimal fraction of the dry soil weight | 0.12 |
| $\rho$ (kg/m$^3$) | bulk density of the soil | 1000 |


**Figures**

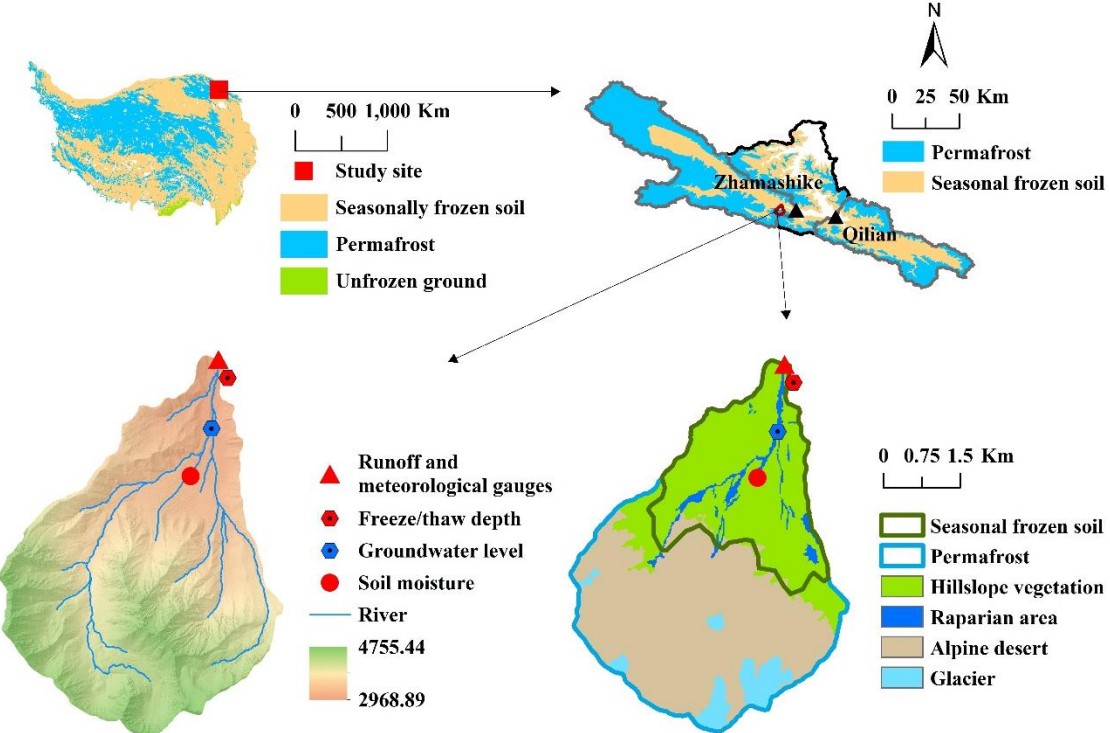


Figure 1. Sketch map of the Qinghai-Tibet Plateau, and the distribution of permafrost and
seasonal frozen-soil of the QTP (Zou et al., 2014), and the location of the upper Heihe River
basin (up left); sketch map of permafrost and seasonal frozen-soil distribution of the upper
Heihe river basin (Sheng, 2020), and the two sub-basins, i.e. Zhamashike and Qilian, and the
location of the Hulu catchment (up right); Hulu catchment's digital elevation model (DEM),
river channel, runoff and meteorological gauge station (observed from 2011 to 2014), the
locations for soil moisture (2011-2013 with data gap), groundwater level (2016-2019), and
freeze/thaw depth (bottom left) (2011-2014); landscapes and seasonal frozen-soil /
permafrost map of the Hulu catchment (bottom right).

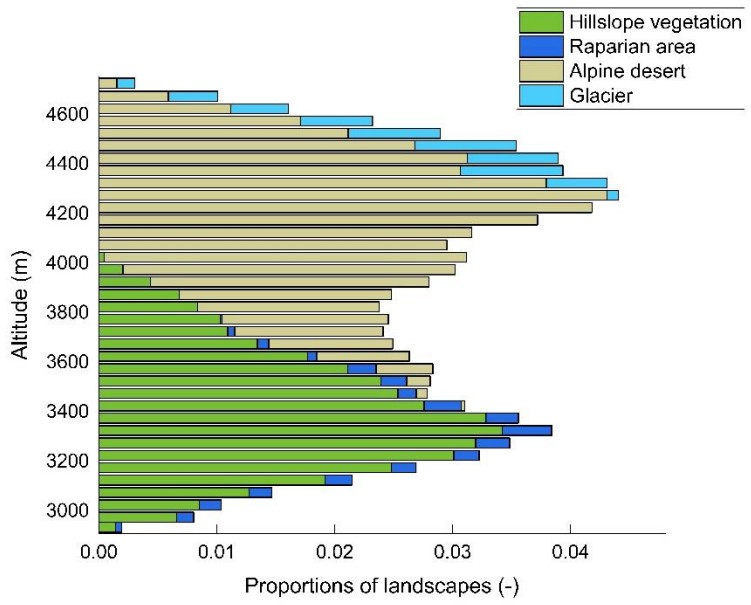


Figure 2. Landscape classification at different elevation bands (with 50m interval) of the Hulu
catchment.

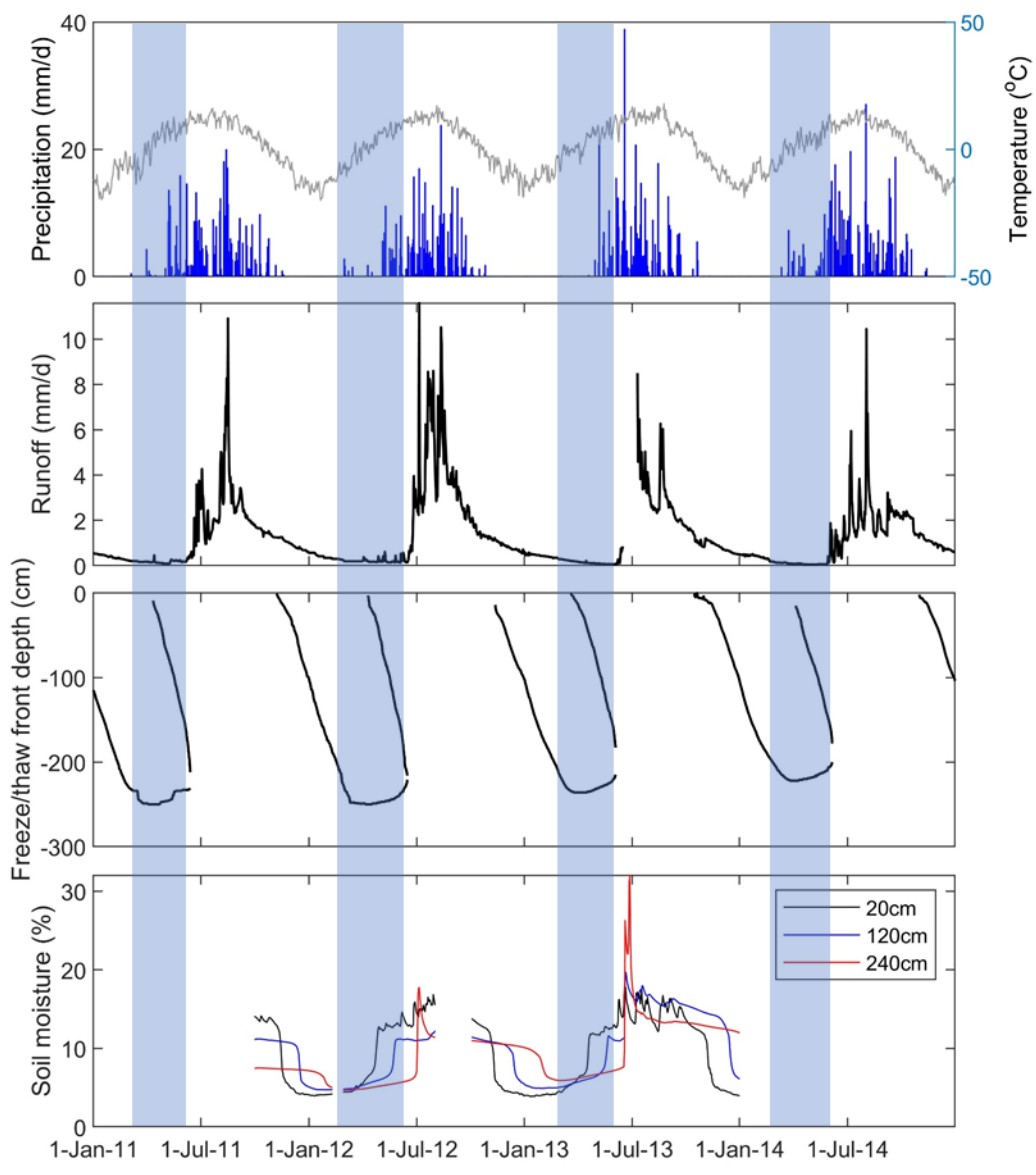


Fig 3. Observed daily precipitation and air temperature; observed daily runoff depth of the
Hulu catchment; observed freeze/thaw front depth; observed soil moisture at the depth of
20cm, 120cm, 240cm.

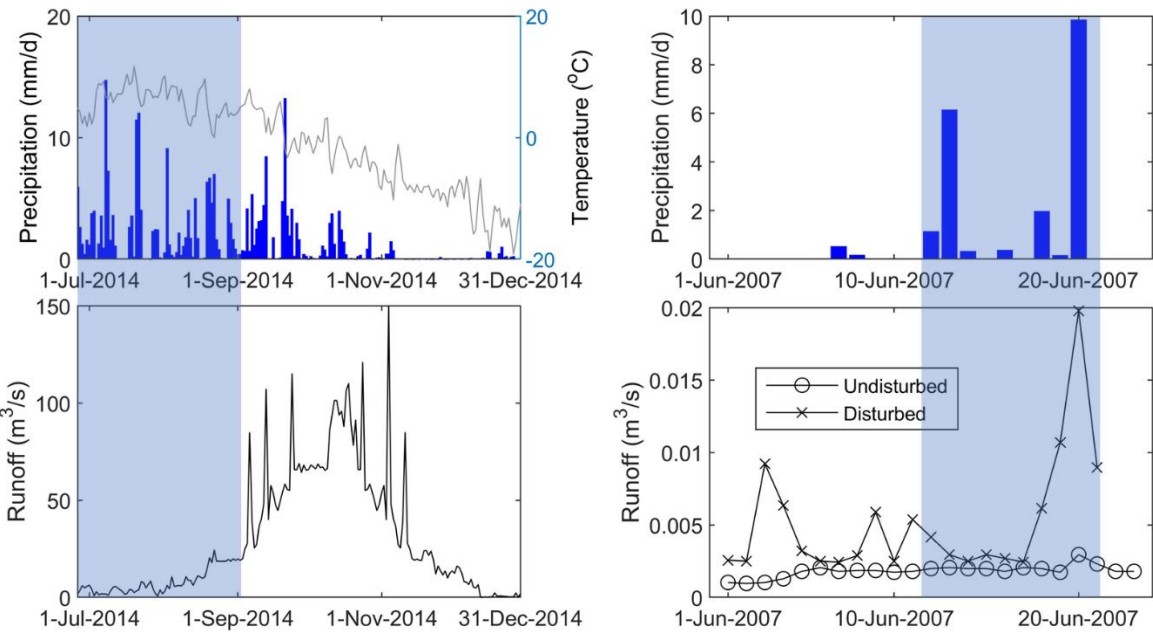


Fig 4. The Little-Runoff in the Early Thawing season (LRET) phenomenon in other places, e.g.
the headwater of Yellow River (Yang et al., 2019), and a small headwater catchment at Cape
Bounty Arctic Watershed Observatory, Melville Island, NU in Canada (Lafrenière and
Lamoureux, 2019)

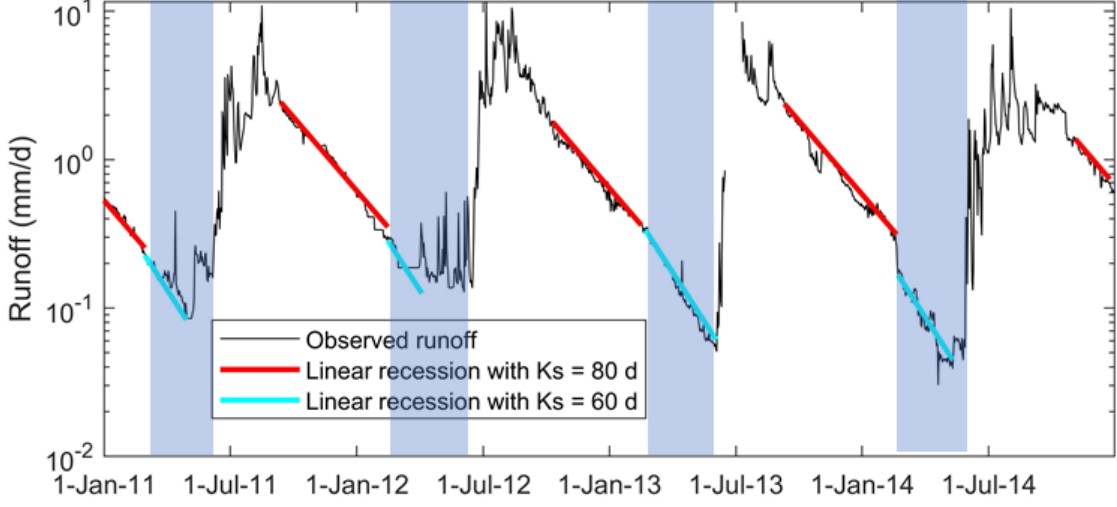


Figure 5. Groundwater recession, from 2011 to 2014, on logarithmic scale, with linear
recession parameter $K_s$ = 80 d in the early recession periods and $K_s$ = 60 d in the end of
recession periods.

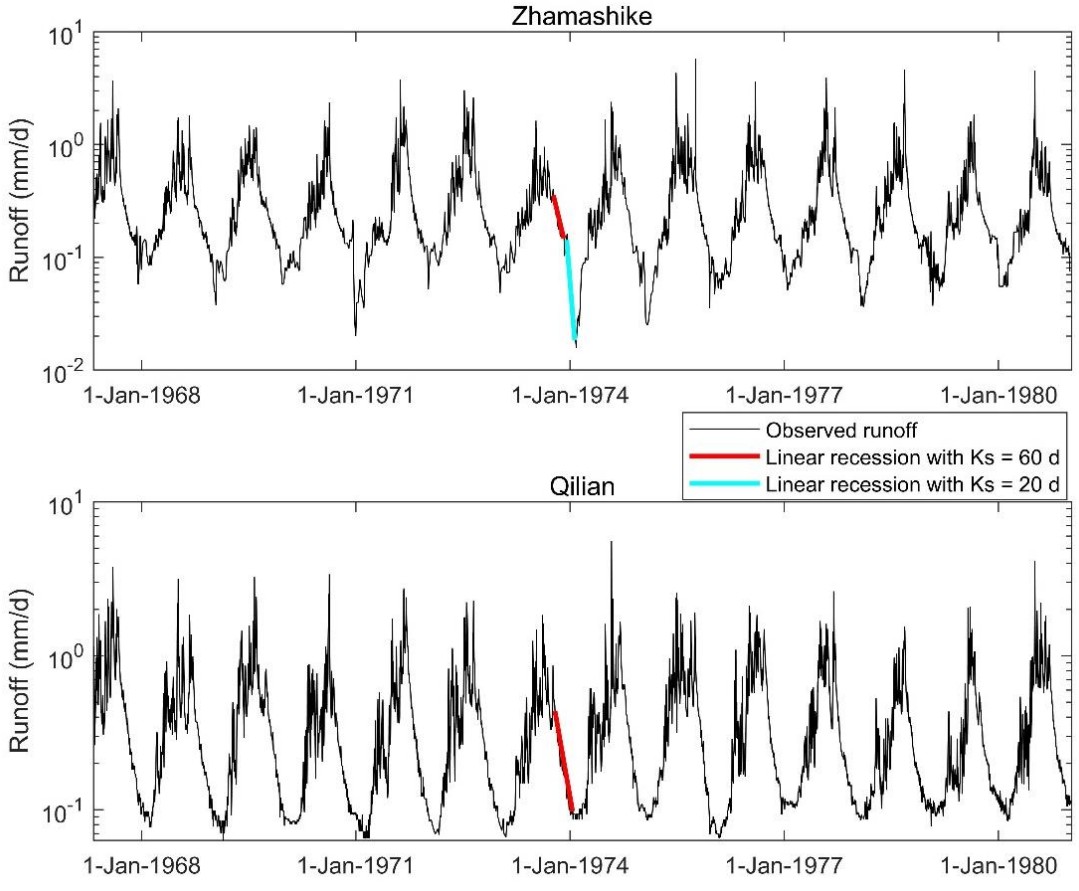


Figure 6. The hydrograph of the Zhamashike and Qilian sub-basin on logarithmic scale, and
the linear recession curve with $K_s$ = 60d and $K_s$ =20d.

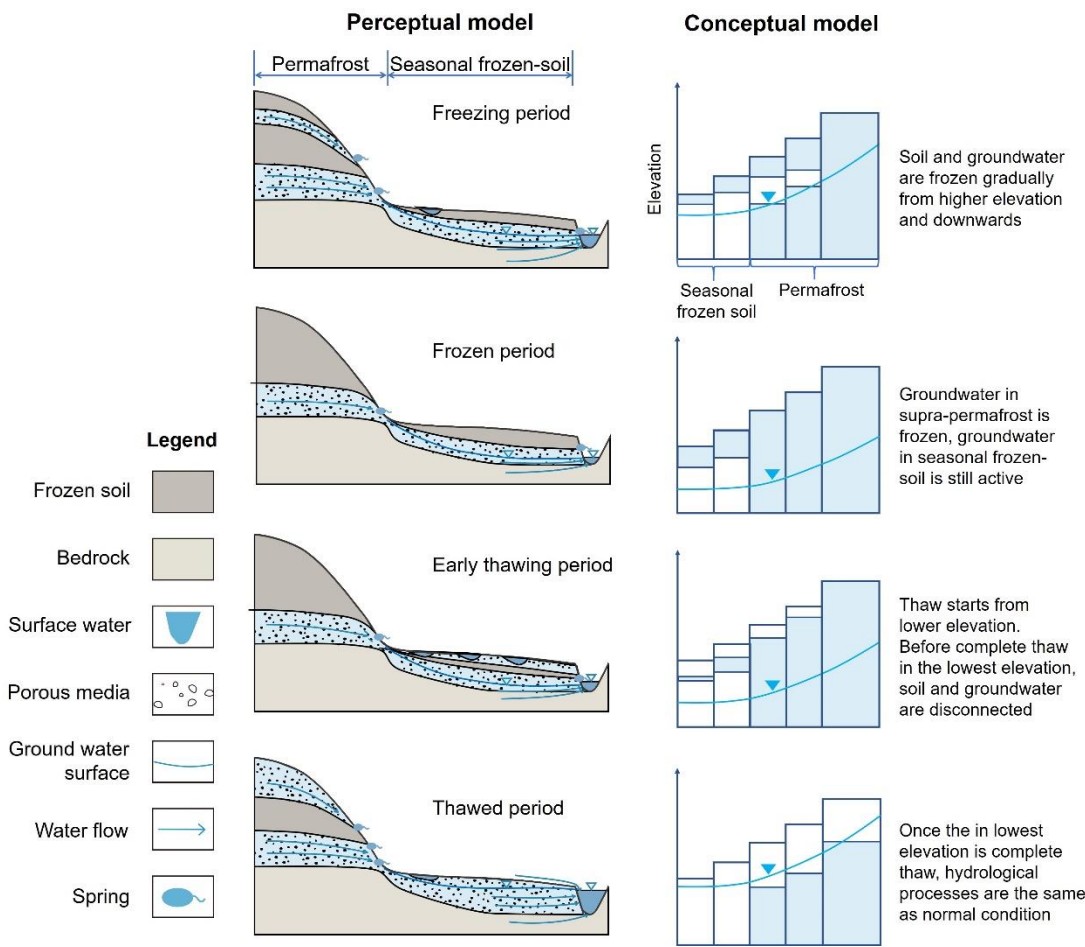


Figure 7 The perceptual and conceptual FLEX-Topo-FS frozen-soil hydrological models.

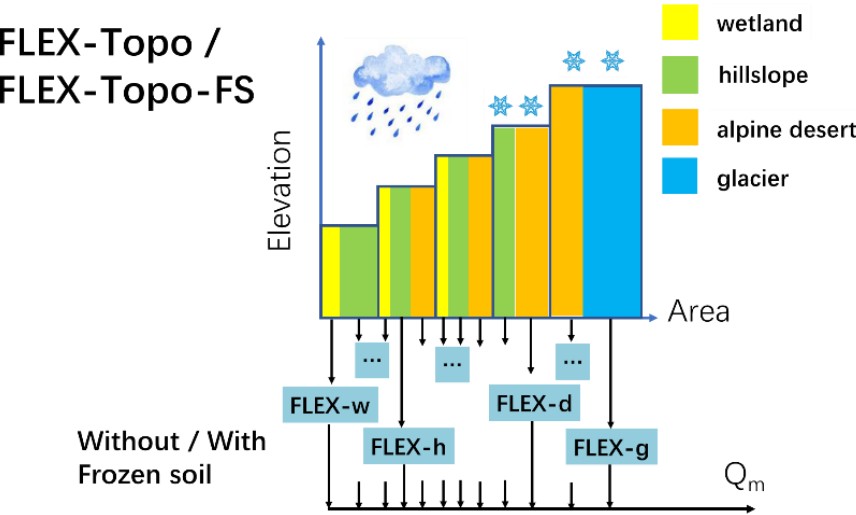


Fig 8. Model structures of FLEX-Topo, and FLEX-Topo-FS. FLEX-w means the module for
wetland, FLEX-h for hillslope, FLEX-d for alpine desert, and FLEX-g for glacier, respectively.

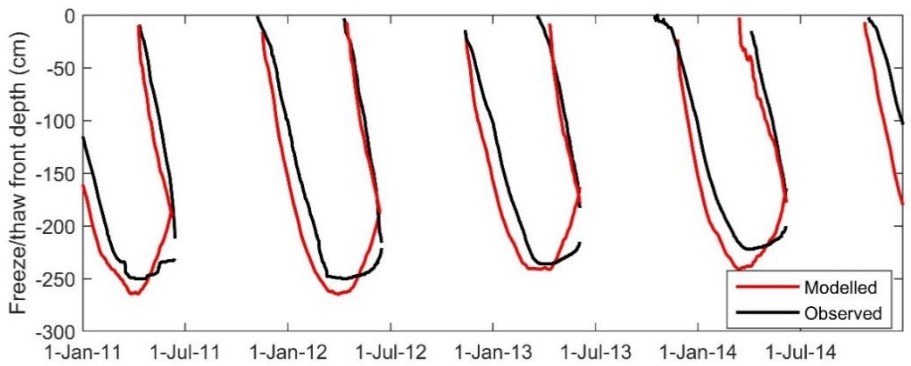


Fig 9. Comparison between simulated freeze/thaw depth by Stefan equation and
observation.

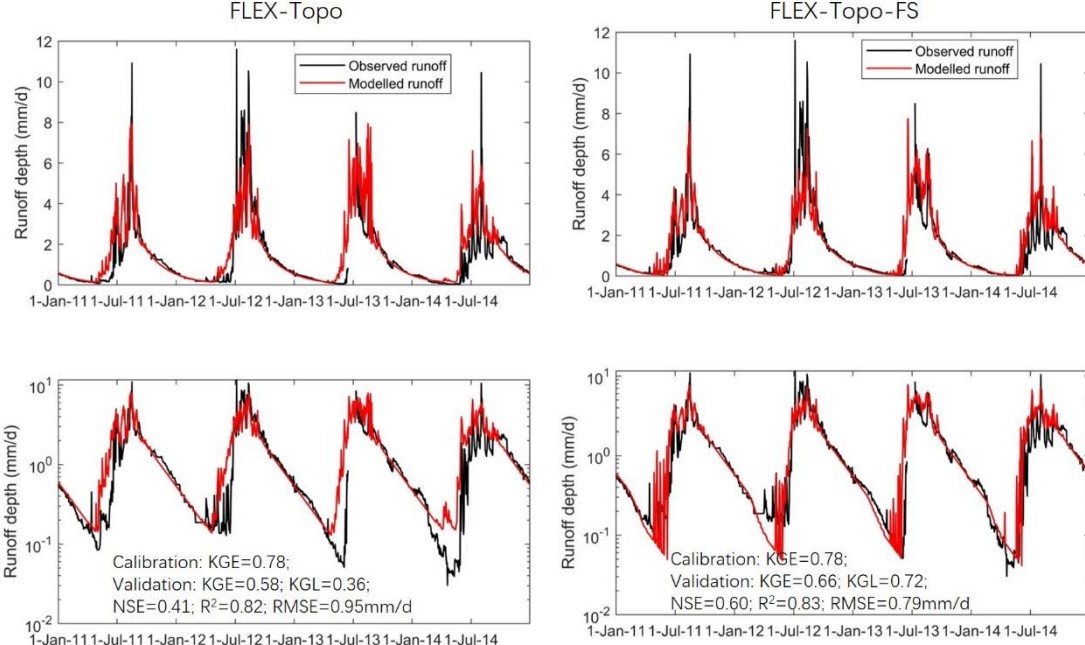


Figure 10. Modeling results of FLEX-Topo and FLEX-Topo-FS models, and the comparisons
with observation, on both normal and logarithm scales.

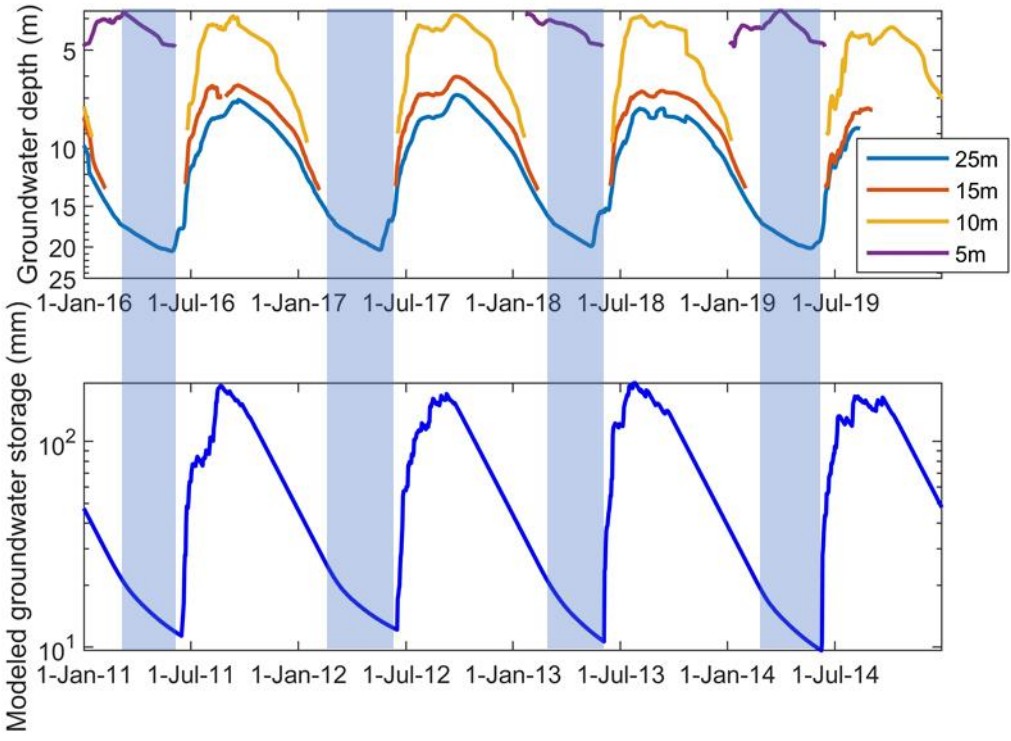

Figure 11. Observed groundwater depth from 2016 to 2019 at WW01 wells at depth of 5m, 10m, 15m, and 25m. And the simulated groundwater storage by the FLEX-Topo-FS model from 2011 to 2014.

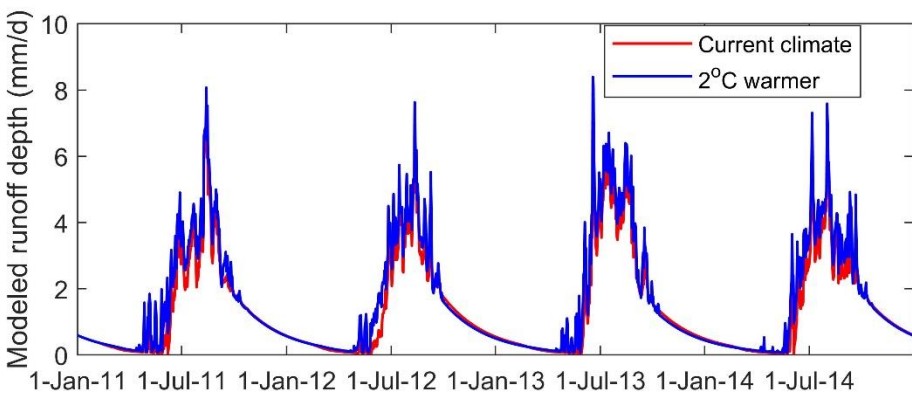

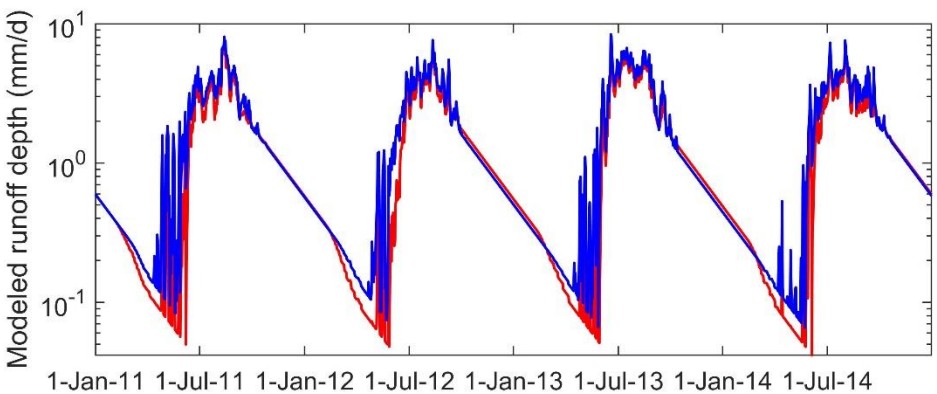

1197

1198 Figure 12. Simulated hydrograph in current climate condition, and the 2°C warmer
1199 condition.