# Peer review of "Frozen-soil hydrological modeling for a mountainous catchment"

_Hydrology and Earth System Sciences, 2022_

## Author Comment (AC1)

The manuscript presents a process of new model development/improvement in frozen ground process. In general, it is clear and reasonable to me. However, the presentation needs proper revisions before it can be considered for publication in HESS. Please clarfiy the merits and limitations of the newly developed frozen ground model, since it is not clear that if this new empirical model can be applied to other cold regions or not.

**Reply**: We thank Referee #1 for the endorsement of the novelty of this manuscript. For the applicability of this model to other cold regions, we believe that the model has great potential to be applied in other cold regions. There are mainly three reasons, which we shall mention in our revised paper.

Firstly, our study site, the Hulu catchment, although small (23 km$^2$), has a large elevation gradient (from 2968m to 4955m), diverse landscapes (hillslope vegetation, riparian area, alpine desert, and glaciers), snowfall and snowmelt, and both permafrost and seasonal frozen-soil. Our newly developed model explicitly considered all these spatial and temporal heterogeneities, and eventually achieved excellent performance. With such a comprehensive modeling toolkit, the model has potential to be upscaled or transfer to other cold regions.

Secondly, we obtained the perceptual model from not only the observations and our expert knowledge at the Hulu catchment itself, but also widely considered the impact of frozen-soil on hydrological processes in other catchments, including the Zhamashike and Qilian (two nested sub-catchments of the upper Heihe), the headwater of Yellow River, and the Cape Bounty Arctic Watershed Observatory in Canada. Thus, we developed the model for the Hulu catchment in the context of larger scale observations.

Thirdly, the realism of the conceptual model was confirmed not only by streamflow measurement, but also by multi-source and multi-scale observations, particularly the freezing and thawing front in the soil, the lower limit of permafrost, and the trends in groundwater level variation.

Although our new model generally has great potential to be used in other cold regions, we should be cautious to arbitrarily use the model without any prior understanding of the modeling system. Since frozen-soil is merely one influential factor for cold region hydrology, there are other factors having notable impacts, which are intertwined with frozen-soil. This relates especially to the geology condition, which can have considerable impact on frozen-soil, but has large spatial heterogeneity, and where it is difficult to take measurements. Hence, before upscaling to other cold regions, we recommend to follow a vigorous modeling procedure, i.e expert-driven data analysis → qualitative perceptual model → quantitative conceptual model → testing of model realism.

Major comments:

(1) It is not really clear if this developed model can be applicable to other river basins. It seems to us that the model development is just from a very small cold river basin of northeast Tibetan Plateau, and the model contains quite a few empirical parameters that needs calibrations with in-situ observations. What is the evidence that this model can be suitable for other basins of Tibetan Plateau?

**Reply**: We thank Referee #1for this question, allowing us to further clarify the modeling details. On model's applicability in other cold regions, which is the same question as your previous one, we refer to the reply above. For the model empirical parameters, most of them are related to the freeze-thaw processes related to Stefan equation, including the thermal conductivity $k$ (W/(m/K)), the water content as a decimal fraction of the dry soil weight $\omega$, the bulk density of the soil $\rho$ (kg/m$^3$), and the multiplier from air temperature to ground temperature during the freezing season. All these parameters have clear physical meanings, confirmed by previous studies with a good spatial distribution over the entire Tibetan Plateau (e.g. Zou et al., 2017; Ran et al., 2022). Due to the extreme complexity of soils in mountainous catchment, we still need to recalibrate their values while modeling other basins on the Tibetan Plateau. So, in general, our answer is Yes. The model has great potential to be used for other cold basins, but we should be cautious to transfer to other catchments, requiring a vigorous test, especially for specific geology conditions.

(2) Figure 1: please clearly describe the time periods of these observations in the figure caption.
**Reply**: We shall do this in the revised manuscript.

(3) Figure 4: Why the runoff is so small in the Arctic watershed? Please also explain the difference between the disturbed and undisturbed.
**Reply**: That is because the runoff was measured in a small Arctic watershed. We will give more explanation in the revised manuscript.

(4) Figure 5: The empirical parameters Ks changes at different periods from 80d to 60d. Is it diffcult to apply/transfer the model to other regions, since you always needs calibrations at different periods?
**Reply**: This is a very good question. Estimating the value of Ks in different catchments and periods is an intriguing scientific question for hydrologists. Brutsaert and Hiyama (2012) studied several Siberian basins ranging from 1,000 km2 to 100,000 km2, and found the Ks was surprisingly in the range of $45\pm15$ days. Brutsaert and Hiyama explained this phenomenon with the theory of catchment self-similarity and co-evolution of geomorphology and hydrology. The discontinuous recession we found in this study is novel. By comparing with other paired catchments, we identified and attributed the discontinuous recession to two periods recession. The first recession period was contributed by the groundwater in both permafrost and seasonal frozen-soil areas, and the second recession period was only contributed by the seasonal frozen-soil area. To our best

knowledge, even without the impacts from frozen-soil, it is difficult to give accurate Ks estimation in small catchments (less than 1,000 km$^2$). The calibration-free Ks in different periods in frozen-soil catchments are more difficult. Thus more studies are still required to further understand the spatial and temporal variation of Ks, not only for frozen-soil basins but also in other climate regions.

(5) Figure 6: Why are the simulated results only shown in one winter at 1974? How about the results in other years? You have never mentioned 1974 in the observations.
**Reply**: Figure 6 shows that discontinuous baseflow recession happened almost every year at the Zhamashike station. We only highlight the hydrograph in 1974 to illustrate the same phenomenon in Zhamashike station as Hulu catchment, since both basins have both permafrost and seasonal frozen-soil, but with very different size (5526 km$^2$ vs 23 km$^2$).

(6) Figure 8: It is not clear here. please explain the FLEX-w, FLEX-h, FLEX-d, FLEX-g in the figure caption.
**Reply**: We will do this in the revised manuscript.

(7) Figure 9, Figure 11 and Figure 10 can be merged. Particularly, Figure 9 and 11 should be put together for direct comparison. Similarly, a new result by FLEX-Topo-FS should be added to compare with Figure 10 that is the simulated result of freeze/thaw depth by FLEX-Topo. For the results in Figure 11, it seems that the new model simulates much more fluctuations that observed. What is the problem about the new model?
**Reply**: This is a good point. We will merge Figure 9 and 11 for direct comparison. For Figure 10, FLEX-Topo did not consider freeze/thaw processes, thus cannot be involved into comparison.

Related to your question about the larger number of fluctuations, we showed the detailed model evaluation metrics in Section 5.2.2: "The KGE of FLEX-Topo-FS in calibration was 0.78, which was the same as FLEX-Topo. But in validation, the performance was significantly improved, the KGE improved from 0.58 to 0.66, KGL was from 0.36 to 0.72, NSE was from 0.41 to 0.60, R$^2$ from 0.82 to 0.83, and RMSE was reduced from 0.95mm/d to 0.79mm/d. All the model evaluation criteria were improved. The most significant improvement was the baseflow simulation, and KGL was increased from 0.36 to 0.72. We also noted that the FLEX-Topo-FS model reproduced the spikes during the thawing in Figure 11. This confirms our conceptual model of a sequence of thawing breakthroughs, which trigger the sudden release of groundwater starting at lower elevations and progressing to higher landscape elements."

(8) Figure 12: the comparison (only showing the periodical variation) is not meaningful, since the time periods for the two graphs are different.
**Reply**: Groundwater fluctuation in natural catchments has strong periodicity, which can be observed in Figure 12. Groundwater variation does not show significant difference

among different years, which is especially true for the 25m well. Due to the extreme difficulty of continuous observation in this region, there was no groundwater measurement in 2011-2014. But due to the strong repeated temporal variation of groundwater level, we have good reason to believe the trends happened in 2016-2019 will also happen in 2011-2014. Moreover, this is a qualitative comparison, rather than a quantitative one, which we don't think is a big problem.

(9) Figure 13: Future projection on 2 degrees warmer is just too simple. Please use IPCC outputs or more scientific designs.

**Reply**: We thank Referee #1 for this suggestion. We also believe it is worthwhile to use IPCC future climate prediction for more robust projection for future changes. In this study, we used 2 degrees warmer more like using a toy model to illustrate how warming will impact on baseflow, to further verify the capability and robustness of the model itself. We agree that using future climate prediction to force model prediction deserves further study.

(10) Reference: This reference list has missed a lot of recent literatures in frozen ground modeling studies at Tibetan Plateau. Regarding this topic, there have been quite a few studies in past five years, at the headwaters of Yangtze, Yellow, Heihe, and other rivers.

**Reply**: We will add more relevant publications in revised manuscript.

**References:**

Brutsaert, W., and T. Hiyama (2012), The determination of permafrost thawing trends from long-term streamflow measurements with an application in eastern Siberia, J. Geophys. Res., 117, D22110, doi:10.1029/2012JD018344.

Ran, Y. Li, X., Cheng, G., Che, J., Aalto, J. Karjalainen, O. Hjort, J., Luoto, M., Jin, H., Obu, J., Hori, M., Yu, Q., Chang, X. (2022) New high-resolution estimates of the permafrost thermal state and hydrothermal conditions over the Northern Hemisphere. Earth System Science Data, 14, 865–884. Doi: 10.5194/essd-14-865-2022.

Zou, D., Zhao, L., Sheng, Y., Chen, J., Hu, G., Wu, T., et al. (2017). A new map of permafrost distribution on the Tibetan Plateau. Cryosphere. 11, 2527–2542. doi:10.5194/tc-11-2527-2017

---

## Author Comment (AC2)

This manuscript aims to develop a conceptual hydrological model for frozen ground. The topic is interesting for hydrological modeling in cold regions. I have some major comments:

**Reply:** We thank Referee #1's endorsement for the contribution of this manuscript. Please find our detailed responses in below.**

1. Section 1.2 need to be improved to explain what are the gaps between our understanding and the real changes in frozen soil. For example, the author mentioned that most current models project a long-term drying of surface soil, dose this projection agree with the real changes? Can the conclusions of this manuscript or the model developed in this manuscript address or explain this problem? What is the contributions for the model developed for improving the prediction of the streamflow in permafrost regions?

**Reply:** We thank Referee's constructive suggestions. We will further improve the Introduction, to highlight the novelty and remove some less relevant narratives.**

2. The method for this manuscript is not easy to follow. The model was developed base on the observations in the small Hulu basin, and then it is validated in the Hulu basin. I suggest to validate the model in other catchment in the upper Heihe basin, the whole upper Heihe basin and other basins in the Qinghai-Tibetan Plateau.

**Reply:** We will revise the methodology part, and clarify more details. This is the same question as Referee #1. And we paste our replies in below:

"For the applicability of this model to other cold regions, we believe that the model has great potential to be applied in other cold regions. There are mainly three reasons, which we shall mention in our revised paper.

Firstly, our study site, the Hulu catchment, although small (23 km2), has a large elevation gradient (from 2968m to 4955m), diverse landscapes (hillslope vegetation, riparian area, alpine desert, and glaciers), snowfall and snowmelt, and both permafrost and seasonal frozen-soil. Our newly developed model explicitly considered all these spatial and temporal heterogeneities, and eventually achieved excellent performance. With such a comprehensive modeling toolkit, the model has potential to be upscaled or transfer to other cold regions.

Secondly, we obtained the perceptual model from not only the observations and our expert knowledge at the Hulu catchment itself, but also widely considered the impact of frozen-soil on hydrological processes in other catchments, including the Zhamashike and Qilian (two nested sub-catchments of the upper Heihe), the headwater of Yellow River,

and the Cape Bounty Arctic Watershed Observatory in Canada. Thus, we developed the model for the Hulu catchment in the context of larger scale observations.

Thirdly, the realism of the conceptual model was confirmed not only by streamflow measurement, but also by multi-source and multi-scale observations, particularly the freezing and thawing front in the soil, the lower limit of permafrost, and the trends in groundwater level variation.

Although our new model generally has great potential to be used in other cold regions, we should be cautious to arbitrarily use the model without any prior understanding of the modeling system. Since frozen-soil is merely one influential factor for cold region hydrology, there are other factors having notable impacts, which are intertwined with frozen-soil. This relates especially to the geology condition, which can have considerable impact on frozen-soil, but has large spatial heterogeneity, and where it is difficult to take measurements. Hence, before upscaling to other cold regions, we recommend to follow a vigorous modeling procedure, i.e expert-driven data analysis  $\rightarrow$  qualitative perceptual model  $\rightarrow$  testing of model realism."

Model validation in the upper Heihe basin and other basins in the Qinghai-Tibetan Plateau is an ambitious model transferability test, which is worthwhile to conduct for further studies, but seems outside the scope of this manuscript.

3. Figure 5, the discontinuous recession seems only evident in 2014. Why?

**Reply:** The discontinuous recession is probably more apparent in 2014, but definitely happened for other three years (2011, 2012, and 2013). What is even more interesting is the spikes during the thawing in Figure 5. This is another evidence for discontinuous recession, which is likely triggered by the sudden release of groundwater starting at lower elevations in the end of frozen seasons. Our model was capable to reproduce the spikes, which further confirmed our conceptual model of a sequence of thawing breakthroughs.

4. Figure 12, in 2 degree warming, the discontinuous recession seems not found, why?

**Reply:** We thank Referee #2 for this valuable question. The current discontinuous recession was caused by permafrost and seasonal frozen-soil. The first recession period was contributed by the groundwater discharge from both permafrost and seasonal frozen-soil areas, and the second recession period was only contributed by the seasonal frozen-soil area. Thus the recession was discontinuous. But two degree warming results in permafrost degradation, and permafrost is degraded to seasonal frozen-soil. Thus there will be only groundwater discharge from seasonal frozen-soil, and lead to continuous baseflow recession. We will add this discussion in the revised manuscript.

5. Section 4.2.2. There are some empirical parameters and settings, such as the 3 m threshold for the frozen depth and 10% for groundwater storage. I suggest to developed more robust equations to represent these processes.

**Reply:** These parameter values are based on our expert knowledge in the field and lab experiments (Romanovsky and Osterkamp, 2000). We agree with the value of further investigating these processes, and our paper is definitely not the end of our exploration in frozen-soil hydrology. In the revised paper we shall clarify this choice.

6. I suggest to show the distributions of HRUs in some figures.

Reply: The HRUs are shown in Figure 1 and 2 in current manuscript.

7. The storage simulated in conceptual models could not be considered as the real "groundwater storage", this should be noted.

**Reply:** Yes, they are different, but they have similar trend, which we believe is a strong model realism test. We will further clarify this in the revision.

**References:**

Romanovsky and Osterkamp. (2000) Effects of Unfrozen Water on Heat and Mass Transport Processes in the Active Layer and Permafrost. Permafrost Periglac. Process. 11: 219-239

---

## Author Response (AR1)

Dear editor and reviewers,

We thank for all your valuable inputs to further improve the quality of this resubmitted manuscript (*Frozen-soil hydrological modeling for a mountainous catchment at northeast of the Tibetan Plateau, HESS-2022-98*), which was a revised version of the previous rejected one (*Diagnosing the impacts of permafrost on catchment hydrology: field measurements and model experiments in a mountainous catchment in western China, hess-2021-264*). In the new round of revision, we received additional constructive comments, which gave us an opportunity to further improve our novel modeling work.

In the revised manuscript, we made significant changes, mostly in two places:

1) We added a new section (Section 6.4), to discuss the potential applicability of our new model in other cold regions.
2) We rephrased part of our perceptual model, to clarify how we modeled the discontinuous baseflow recession (DBR).

To the best of our knowledge, this study is the first report of LRET and DBR processes in a mountainous frozen-soil catchment. In addition, the FLEX-Topo-FS model is a novel conceptual for frozen-soil hydrological modeling, which is one of the twenty-three unsolved hydrology problems (23UHP). Please found our detailed point-by-point responses in below.

Your sincerely,

Hongkai Gao, on behalf of all co-authors

**Anonymous Referee #1**

The manuscript presents a process of new model development/improvement in frozen ground process. In general, it is clear and reasonable to me. However, the presentation needs proper revisions before it can be considered for publication in HESS. Please clarfiy the merits and limitations of the newly developed frozen ground model, since it is not clear that if this new empirical model can be applied to other cold regions or not.

**Reply**: We thank Referee #1 for the endorsement of the novelty of this manuscript. For the applicability of this model to other cold regions, we believe that the model has great potential to be applied in other cold regions. There are mainly three reasons, which we mentioned in our revised paper (see Section 6.4).

Firstly, our study site, the Hulu catchment, although small (23 km$^2$), has a large elevation gradient (from 2968m to 4955m), diverse landscapes (hillslope vegetation, riparian area,

alpine desert, and glaciers), snowfall and snowmelt, and both permafrost and seasonal frozen-soil. Our newly developed model explicitly considered all these spatial and temporal heterogeneities, and eventually achieved excellent performance. With such a comprehensive modeling toolkit, the model has potential to be upscaled or transfer to other cold regions.

Secondly, we obtained the perceptual model from not only the observations and our expert knowledge at the Hulu catchment itself, but also widely considered the impact of frozen-soil on hydrological processes in other catchments, including the Zhamashike and Qilian (two nested sub-catchments of the upper Heihe), the headwater of Yellow River, and the Cape Bounty Arctic Watershed Observatory in Canada. Thus, we developed the model for the Hulu catchment in the context of larger scale observations.

Thirdly, the realism of the conceptual model was confirmed not only by streamflow measurement, but also by multi-source and multi-scale observations, particularly the freezing and thawing front in the soil, the lower limit of permafrost, and the trends in groundwater level variation.

Although our new model generally has great potential to be used in other cold regions, we should be cautious to arbitrarily use the model without any prior understanding of the modeling system. Since frozen-soil is merely one influential factor for cold region hydrology, there are other factors having notable impacts, which are intertwined with frozen-soil. This relates especially to the geology condition, which can have considerable impact on frozen-soil, but has large spatial heterogeneity, and where it is difficult to take measurements. Hence, before upscaling to other cold regions, we recommend to follow a stringent modeling procedure, i.e expert-driven data analysis → qualitative perceptual model → quantitative conceptual model → testing of model realism.

Major comments:

(1) It is not really clear if this developed model can be applicable to other river basins. It seems to us that the model development is just from a very small cold river basin of northeast Tibetan Plateau, and the model contains quite a few empirical parameters that needs calibrations with in-situ observations. What is the evidence that this model can be suitable for other basins of Tibetan Plateau?

**Reply**: We thank Referee #1for this question, allowing us to further clarify the modeling details. For the empirical parameters, most of them are related to the freeze-thaw processes related to Stefan equation, including the thermal conductivity $k$ (W/(m/K)), the water content as a decimal fraction of the dry soil weight $\omega$, the bulk density of the soil $\rho$ (kg/m$^3$), and the multiplier from air temperature to ground temperature during the freezing season. All these parameters have clear physical meaning, confirmed by previous studies with a good spatial distribution over the entire Tibetan Plateau (e.g. Zou et al., 2017; Ran et al., 2022). Due to the extreme complexity of soils in mountainous catchments, we still need to recalibrate their values while modeling other basins on the

Tibetan Plateau. So, in general, our answer is Yes. The model has great potential to be used for other cold basins, but we should be cautious to transfer to other catchments, requiring a stringent test, especially for specific geology conditions.

On model's applicability in other cold regions, which is the same question as your previous one, we presented three reasons why we consider it applicable in Section 6.4 in the revised manuscript, and summarized as a reply to point 2 of Referee#2.

(2) Figure 1: please clearly describe the time periods of these observations in the figure caption.
**Reply**: We did this in the revised figure caption.

(3) Figure 4: Why the runoff is so small in the Arctic watershed? Please also explain the difference between the disturbed and undisturbed.
**Reply**: We clarified in Section 3.1.

(4) Figure 5: The empirical parameters Ks changes at different periods from 80d to 60d. Is it diffcult to apply/transfer the model to other regions, since you always needs calibrations at different periods?
**Reply**: This is a very good question. Estimating the value of Ks in different catchments and periods is an intriguing scientific question for hydrologists. Brutsaert and Hiyama (2012) studied several Siberian basins ranging from 1,000 km2 to 100,000 km2, and found that Ks varied in the relatively narrow the range of $45 \pm 15$ days, suggesting potential for parameter transferability. Brutsaert and Hiyama interpreted such consistency of parameter values, advocating the principle of catchment self-similarity and the co-evolution of geomorphology and hydrology. The discontinuous recession we found in this study is novel. By comparing with other paired catchments, we identified and attributed the discontinuous recession to two periods recession. The first recession period was contributed by the groundwater in both permafrost and seasonal frozen-soil areas, and the second recession period was only contributed by the seasonal frozen-soil area. To the best of our knowledge, even without the impacts from frozen-soil, it is difficult to give accurate Ks estimation in small catchments (less than 1,000 km$^2$). The calibration-free Ks in different periods in frozen-soil catchments are more difficult. Thus more studies are still required to further understand the spatial and temporal variation of Ks, not only for frozen-soil basins but also in other climate regions.

We did more discussion in Section 3.3 and 6.1.1.

(5) Figure 6: Why are the simulated results only shown in one winter at 1974? How about the results in other years? You have never mentioned 1974 in the observations.
**Reply**: Figure 6 shows that discontinuous baseflow recession happened almost every year

at the Zhamashike station. We only highlight the hydrograph in 1974 to illustrate the same phenomenon in Zhamashike station as Hulu catchment, since both basins have both permafrost and seasonal frozen-soil, but with very different size (5526 km$^2$ vs 23 km$^2$).

We clarified this choice in Section 3.2.

(6) Figure 8: It is not clear here. please explain the FLEX-w, FLEX-h, FLEX-d, FLEX-g in the figure caption.
**Reply**: We did this in the revised manuscript.

(7) Figure 9, Figure 11 and Figure 10 can be merged. Particularly, Figure 9 and 11 should be put together for direct comparison. Similarly, a new result by FLEX-Topo-FS should be added to compare with Figure 10 that is the simulated result of freeze/thaw depth by FLEX-Topo. For the results in Figure 11, it seems that the new model simulates much more fluctuations that observed. What is the problem about the new model?
**Reply**: This is a good point. We merged Figure 9 and 11 for direct comparison in the revised manuscript. For Figure 10, FLEX-Topo did not consider freeze/thaw processes, thus cannot be involved into comparison.

Related to the question about the larger number of fluctuations, we showed the detailed model evaluation metrics in Section 5.2.2: "The KGE of FLEX-Topo-FS in calibration was 0.78, which was the same as FLEX-Topo. But in validation, the performance was significantly improved, the KGE improved from 0.58 to 0.66, KGL was from 0.36 to 0.72, NSE was from 0.41 to 0.60, R$^2$ from 0.82 to 0.83, and RMSE was reduced from 0.95mm/d to 0.79mm/d. All the model evaluation criteria were improved. The most significant improvement was the baseflow simulation, and KGL was increased from 0.36 to 0.72. We also noted that the FLEX-Topo-FS model reproduced the spikes during the thawing in Figure 11. This confirms our conceptual model of a sequence of thawing breakthroughs, which trigger the sudden release of groundwater starting at lower elevations and progressing to higher landscape elements."

(8) Figure 12: the comparison (only showing the periodical variation) is not meaningful, since the time periods for the two graphs are different.
**Reply**: Groundwater fluctuation in natural catchments has strong periodicity, which can be observed in Figure 12. Groundwater variation does not show significant difference among different years. Due to the extreme difficulty of continuous observation in this region, there was no groundwater measurement in 2011-2014. But due to the strong repeated temporal variation of groundwater level, we have good reason to believe the trends happened in 2016-2019 have also happened in 2011-2014. Moreover, this is intended as a qualitative comparison, rather than a quantitative one, in order to show consistency of behavior. Thus, we do not think this will impact on our main conclusions.

We did more discussion in Section 6.1.2.

(9) Figure 13: Future projection on 2 degrees warmer is just too simple. Please use IPCC outputs or more scientific designs.

**Reply**: We thank Referee #1 for this suggestion. We also believe it is worthwhile to use IPCC future climate prediction for more robust projection for future changes. In this study, we used 2 degrees warmer as a sensitivity analysis to illustrate how warming will impact on baseflow, to further verify the capability and robustness of the model itself. We agree that using future climate prediction to force model prediction deserves further study.

We did more clarification in Section 6.3.

(10) Reference: This reference list has missed a lot of recent literatures in frozen ground modeling studies at Tibetan Plateau. Regarding this topic, there have been quite a few studies in past five years, at the headwaters of Yangtze, Yellow, Heihe, and other rivers.

**Reply**: We added more relevant references in revised manuscript.

**References:**

Brutsaert, W., and T. Hiyama (2012), The determination of permafrost thawing trends from long-term streamflow measurements with an application in eastern Siberia, J. Geophys. Res., 117, D22110, doi:10.1029/2012JD018344.

Ran, Y. Li, X., Cheng, G., Che, J., Aalto, J. Karjalainen, O. Hjort, J., Luoto, M., Jin, H., Obu, J., Hori, M., Yu, Q., Chang, X. (2022) New high-resolution estimates of the permafrost thermal state and hydrothermal conditions over the Northern Hemisphere. Earth System Science Data, 14, 865–884. Doi: 10.5194/essd-14-865-2022.

Song C, Wang G, Mao T, Dai J, Yang D. 2020. Linkage between permafrost distribution and river runoff changes across the Arctic and the Tibetan Plateau. Science China Earth Sciences, 63: 292–302

Zou, D., Zhao, L., Sheng, Y., Chen, J., Hu, G., Wu, T., et al. (2017). A new map of permafrost distribution on the Tibetan Plateau. Cryosphere. 11, 2527–2542. doi:10.5194/tc-11-2527-2017

**Anonymous Referee #2**

This manuscript aims to develop a conceptual hydrological model for frozen ground. The topic is interesting for hydrological modeling in cold regions. I have some major comments:

**Reply:** We thank Referee #1's endorsement for the contribution of this manuscript. Please find our detailed responses in below.

1. Section 1.2 need to be improved to explain what are the gaps between our understanding and the real changes in frozen soil. For example, the author mentioned that most current models project a long-term drying of surface soil, dose this projection agree with the real changes? Can the conclusions of this manuscript or the model developed in this manuscript address or explain this problem? What is the contributions for the model developed for improving the prediction of the streamflow in permafrost regions?

**Reply:** We thank Referee's constructive suggestions.

We further improved the Introduction, to highlight the novelty and remove some less relevant narratives.

2. The method for this manuscript is not easy to follow. The model was developed base on the observations in the small Hulu basin, and then it is validated in the Hulu basin. I suggest to validate the model in other catchment in the upper Heihe basin, the whole upper Heihe basin and other basins in the Qinghai-Tibetan Plateau.

**Reply:** We revised the methodology part, and clarified with more details. This is the same question as Referee #1.

For the applicability of this model to other cold regions, we believe that the model has great potential to be applied in other cold regions. There are mainly three reasons, which we mentioned in our revised paper (see Section 6.4).

Firstly, our study site, the Hulu catchment, although small (23 km$^2$), has a large elevation gradient (from 2968m to 4955m), diverse landscapes (hillslope vegetation, riparian area, alpine desert, and glaciers), snowfall and snowmelt, and both permafrost and seasonal frozen-soil. Our newly developed model explicitly considered all these spatial and temporal heterogeneities, and eventually achieved excellent performance. With such a comprehensive modeling toolkit, the model has potential to be upscaled or transfer to other cold regions.

Secondly, we obtained the perceptual model from not only the observations and our expert knowledge at the Hulu catchment itself, but also widely considered the impact of frozen-soil on hydrological processes in other catchments, including the Zhamashike and

Qilian (two nested sub-catchments of the upper Heihe), the headwater of Yellow River, and the Cape Bounty Arctic Watershed Observatory in Canada. Thus, we developed the model for the Hulu catchment in the context of larger scale observations.

Thirdly, the realism of the conceptual model was confirmed not only by streamflow measurement, but also by multi-source and multi-scale observations, particularly the freezing and thawing front in the soil, the lower limit of permafrost, and the trends in groundwater level variation.

Although our new model generally has great potential to be used in other cold regions, we should be cautious to arbitrarily use the model without any prior understanding of the modeling system. Since frozen-soil is merely one influential factor for cold region hydrology, there are other factors having notable impacts, which are intertwined with frozen-soil. This relates especially to the geology condition, which can have considerable impact on frozen-soil, but has large spatial heterogeneity, and where it is difficult to take measurements. Hence, before upscaling to other cold regions, we recommend to follow a stringent modeling procedure, i.e expert-driven data analysis → qualitative perceptual model → quantitative conceptual model → testing of model realism."

Model validation in the upper Heihe basin and other basins in the Qinghai-Tibetan Plateau is an ambitious model transferability test, which is worthwhile to conduct for further studies, but seems outside the scope of this manuscript.

3. Figure 5, the discontinuous recession seems only evident in 2014. Why?

**Reply:** The discontinuous recession is probably more apparent in 2014, but definitely happened for other three years (2011, 2012, and 2013). What is even more interesting is the spikes during the thawing in Figure 5. This is another evidence for discontinuous recession, which is likely triggered by the sudden release of groundwater starting at lower elevations in the end of frozen seasons. Our model was capable to reproduce the spikes, which further confirmed our conceptual model of a sequence of thawing breakthroughs.

We did more discussion in Section 3.2 and 3.3.

4. Figure 12, in 2 degree warming, the discontinuous recession seems not found, why?

**Reply:** We thank Referee #2 for this valuable question. The current discontinuous recession was caused by the different groundwater behaviors in permafrost and seasonal frozen-soil areas. The first recession period was contributed by the groundwater discharge from both permafrost and seasonal frozen-soil areas, and the second recession period was only contributed by the seasonal frozen-soil area. Thus the recession was discontinuous. But two-degree warming results in permafrost degradation, and permafrost is degraded to seasonal frozen-soil. Thus there will be mostly groundwater discharge from seasonal frozen-soil, and lead to continuous baseflow recession.

We added more discussion in Section 6.3.

5. Section 4.2.2. There are some empirical parameters and settings, such as the 3 m threshold for the frozen depth and 10% for groundwater storage. I suggest to developed more robust equations to represent these processes.

**Reply:** These parameter values are based on our expert knowledge in the field and lab experiments (Romanovsky and Osterkamp, 2000). We agree with the value of further investigating these processes, and our paper is definitely not the end of our exploration in frozen-soil hydrology.

In the revised paper we did clarify this in 4.2.2.

6. I suggest to show the distributions of HRUs in some figures.

**Reply:** The HRUs are shown in Figure 1 and 2 in the manuscript.

7. The storage simulated in conceptual models could not be considered as the real "groundwater storage", this should be noted.

**Reply:** Yes, they are different, but they have similar trend, which we believe is a strong model realism test.

We added more clarification in Section 5.2.3.

**References:**

Romanovsky and Osterkamp. (2000) Effects of Unfrozen Water on Heat and Mass Transport Processes in the Active Layer and Permafrost. Permafrost Periglac. Process. 11: 219-239

---

## Author Response (AR2)

**Anonymous referee #1:**

The authors have generally addressed my previous comments. It can be accepted for publicaiton in HESS.

Reply: We thank Referee #1's endorsement for the acceptance for publication in HESS.

**Anonymous referee #2:**

The author has answered all of my questions, and some revisions are adequate. However, I still has concerns about to develop a robust model. My comments are shown bellow:

Reply: We are glad that Referee #2' satisfied with our revision. Please find our answers for your detailed questions.

1. The author mentioned that the perceptual model comes from the observations from not only the observations and expert knowledges at the Hulu catchment itself, but also widely considered the impact of frozen-soil on hydrological processes in other catchments, including the Zhamashike and Qilian (two nested sub-catchments of the upper Heihe). Therefore, I think the model can be easy to apply at the nested Zhamashike and Qilian catchment and also the whole upper Heihe basin. To validate the model performance in the Zhamashike and Qilian catchment or the whole basin is necessary to prove the model's accuracy. This is because spatial scale problems for hydrology usually exist although there is spatial variations of landscape in the small Hulu catchment. As scale increasing, the response of hydrology processes to atmosphere forcings may be different.

   Reply: We also believe that transferring the frozen-soil hydrological model from Hulu catchment to larger upper Heihe basin and its sub-basins is a stronger model realism test. We will conduct this study in the near future. Since the length of current manuscript is already over 13.8k words, and doing model transferability test needs more detailed landscape analysis and meteorological forcing distribution, which are out the scope of this paper. We believe current multi-source, multi-scale and multi-catchment measurements are strong enough to support the realism of our model, including not only the hydrography measurements in multi-catchments with different permafrost coverage, but also multi-source and multi-scale observations, particularly the freezing and thawing front in the soil, the lower limit of permafrost, and the trends in groundwater level variation. We thank Referee #2's valuable inputs which are valuable to guide our future studies. This paper is definitely not the end of our exploration in frozen-soil hydrology.

2. The author mentioned that the model empirical parameters, most of them are related to the freeze- thaw processes related to Stefan equation. I think it need more explanation for the relationship between the Stefan equation and all the parameters in table 2.

   Reply: This is a good question. Stefan equation described the soil freeze/thaw processes, which are mostly the vertical processes. The parameters in Table 2 are all about catchment hydrological processes, which involved both vertical and lateral processes. All this paper is to link soil freeze/thaw processes with catchment hydrology in a process-based way. The detailed methodology for linking soil freeze/thaw processes (calculated by Stefan equation) and hydrological parameters can be found mostly in Section 4.2.2.